# Online Optimization with Memory and Competitive Control

**Guanya Shi\***
CMS, Caltech
gshi@caltech.edu

**Yiheng Lin\***
IIIS, Tsinghua University
linyh16@mails.tsinghua.edu.cn

**Soon-Jo Chung**
CMS, GALCIT, JPL, Caltech
sjchung@caltech.edu

**Yisong Yue**
CMS, Caltech
yyue@caltech.edu

**Adam Wierman**
CMS, Caltech
adamw@caltech.edu

## Abstract

This paper presents competitive algorithms for a novel class of online optimization problems with memory. We consider a setting where the learner seeks to minimize the sum of a hitting cost and a switching cost that depends on the previous $p$ decisions. This setting generalizes Smoothed Online Convex Optimization. The proposed approach, Optimistic Regularized Online Balanced Descent, achieves a constant, dimension-free competitive ratio. Further, we show a connection between online optimization with memory and online control with adversarial disturbances. This connection, in turn, leads to a new constant-competitive policy for a rich class of online control problems.

## 1 Introduction

This paper studies the problem of Online Convex Optimization (OCO) with *memory*, a variant of classical OCO [25] where an online learner iteratively picks an action $y_t$ and then suffers a convex loss $g_t(y_{t-p}, \cdots, y_t)$, depending on current and *previous* actions. Incorporating memory into OCO has seen increased attention recently, due to both its theoretical implications, such as to convex body chasing problems [11, 8, 37, 12], and its wide applicability to settings such as data centers [32], power systems [30, 9, 26], and electric vehicle charging [26, 17]. Of particular relevance to this paper is the considerable recent effort studying connections between OCO with memory with online control in dynamical systems, leading to online algorithms that enjoy sublinear static regret [4, 5], low dynamic regret [29, 31], constant competitive ratio [23], and the ability to boost weak controllers [3].

Prior work on OCO with memory is typically limited in one of two ways. First, algorithms with the strongest guarantees, a constant *competitive ratio*, are restricted to a special case known as Smoothed Online Convex Optimization (SOCO), or OCO with switching costs [16, 32, 24], which considers only one step of memory and assumes cost functions can be observed before actions are chosen. Second, algorithms proposed for the general case typically only enjoy sublinear *static regret* [6], which is a much weaker guarantee, because static regret compares to the offline optimal static solution while competitive ratio directly compares to the true offline optimal. It is known that algorithms that achieve sublinear static regret can be arbitrarily worse than the true offline optimal [22], and also may have unbounded competitive ratios [7]. The pursuit of general-purpose constant-competitive algorithms for OCO with memory remains open.

Our work is also motivated by establishing theoretical connections between online optimization and control. Recently a line of work has shown the applicability of tools from online optimization

---

for control, albeit in limited settings [4, 5, 28, 23]. Deepening these connections can potentially be impactful since most prior work studies how to achieve sublinear regret compared to the best static linear controller [20, 5, 4, 19]. However, the best static linear controller is a weak benchmark compared to the true optimal controller [22], which may be neither linear nor static. To achieve stronger guarantees, one must seek to bound either the competitive ratio [23] or dynamic regret [29, 31], and connections to online optimization can provide such results. However, prior attempts either have significant caveats (e.g., bounds depend on the path length of the instance [29, 31]) or only apply to very restricted control systems (e.g., invertible control actuation matrices and perfect knowledge of disturbances [23]). As such, the potential to obtain constant-competitive policies for general control systems via online optimization remains unrealized.

**Main contributions.** We partially bridge the two gaps highlighted above. First, we propose a novel setting, OCO with *structured* memory, where the cost function depends on the previous $p$ decisions and is not known precisely before determining the action. This setting generalizes SOCO to include more than one step of memory and to eliminate the assumption that the cost function must be perfectly known before choosing the action. Second, we propose a novel algorithm, Optimistic Regularized Online Balanced Descent, that has a constant and dimension-free competitive ratio for OCO with structured memory. This is the first algorithm with a constant competitive ratio for online optimization with memory longer than one step. Third, we provide a nontrivial reduction from a rich class of online control problems to OCO with structured memory and, via the reduction, show that a constant-competitive policy exists for this class of control problems. While not completely general, the class of problems is considerably more general than existing settings where competitive polices are known, e.g., the control matrix must be invertible and the disturbances are known in advance [23]. Finally, we use examples to (i) demonstrate the gap between the best offline linear policy and the true optimal offline policy can be arbitrarily large, and (ii) highlight that our algorithms can significantly outperform the best offline linear controller, which serves as the benchmark of no-regret policies.

## 2 Background and model

In this section, we formally present the problem setting in this paper. We first survey prior work on OCO with memory and then introduce our new model of OCO with structured memory. Throughout this paper, $M_{i:j}$ denotes either $\{M_i, M_{i+1}, \cdots, M_j\}$ if $i \leq j$, or $\{M_i, M_{i-1}, \cdots, M_j\}$ if $i > j$.

### 2.1 Online convex optimization with memory

Online convex optimization (OCO) with memory is a variation of classical OCO that was first introduced by Anava et al. [6]. In contrast to classical OCO, in OCO with memory, the loss function depends on previous actions in addition to the current action. At time step $t$, the online agent picks $y_t \in \mathcal{K} \subset \mathbb{R}^d$ and then a loss function $g_t : \mathcal{K}^{p+1} \to \mathbb{R}$ is revealed. The agent incurs a loss of $g_t(y_{t-p:t})$. Thus, $p$ quantifies the length of the memory in the loss function. Within this general model of OCO with memory, Anava et al. [6] focus on developing policies with small *policy regret*, which is defined as:

$$\texttt{PolicyRegret} = \sum_{t=p}^{T} g_t(y_{t-p:t}) - \min_{y \in \mathcal{K}} \sum_{t=0}^{T} g_t(y, \cdots, y).$$

The main result presents a memory-based online gradient descent algorithm that achieves $O(\sqrt{T})$ regret under some moderate assumptions on the diameter of $\mathcal{K}$ and the gradient of the loss functions.

**Online convex optimization with switching costs (SOCO).** While the general form of OCO with memory was introduced only recently, specific forms of OCO problems involving memory have been studied for decades. Perhaps the most prominent example is OCO with switching costs, often termed Smoothed Online Convex Optimization (SOCO) [32, 14, 16, 23, 30, 24]. In SOCO, the loss function is separated into two pieces: (i) a *hitting cost* $f_t$, which depends on only the current action $y_t$, and a *switching cost* $c(y_t, y_{t-1})$, which penalizes big changes in the action between rounds. Often the hitting cost is assumed to be of the form $\|y_t - v_t\|$ for some (squared) norm, motivated by tracking some unknown trajectory $v_t$, and the switching cost $c$ is a (squared) norm motivated by penalizing switching in proportion to the (squared) distance between the actions, e.g., a common choice $c(y_t, y_{t-1}) = \frac{1}{2}\|y_t - y_{t-1}\|_2^2$ [23, 30]. The goal of the online learner is to minimize its total cost over $T$ rounds: $\texttt{cost}(ALG) = \sum_{t=1}^{T} f_t(y_t) + c(y_t, y_{t-1})$.

Under SOCO, results characterizing the policy regret are straightforward, and the goal is instead to obtain stronger results that characterize the *competitive ratio*. The competitive ratio is the worst-case ratio of total cost incurred by the online learner and the offline optimal. The cost of the offline optimal is defined as the minimal cost of an algorithm if it has full knowledge of the sequence $\{f_t\}$, i.e.: $\texttt{cost}(OPT) = \min_{y_1 \dots y_T} \sum_{t=1}^{T} f_t(y_t) + c(y_t, y_{t-1})$. Using this, the *competitive ratio* is defined as:

$$\texttt{CompetitiveRatio}(ALG) = \sup_{f_{1:T}} \frac{\texttt{cost}(ALG)}{\texttt{cost}(OPT)}.$$

Bounds for competitive ratio are stronger than for policy regret, since the dynamic offline optimal can change its decisions on different time steps [6].

In the context of SOCO, the first results bounding the competitive ratio focused on one-dimensional action sets [33, 10], but after a long series of papers there now exist algorithms that provide constant competitive ratios in high dimensional settings [16, 23, 24]. Among different choices of switching cost $c$, we are particularly interested in $c(y_t, y_{t-1}) = \frac{1}{2} \|y_t - y_{t-1}\|_2^2$ due to the connection to quadratic costs in control problems. The state-of-the-art algorithm for this switching cost is Regularized Online Balanced Descent (ROBD), introduced by Goel et al. [24], which achieves the lowest possible competitive ratio of any online algorithm. Other recent results study the case where $c(y_t, y_{t-1}) = \|y_t - y_{t-1}\|$ [11, 8, 37, 12]. Variations of the problem with predictions [14, 15, 30], non-convex cost functions [35], and constraints [34, 40] have been studied as well.

## 2.2 OCO with structured memory

Though competitive algorithms have been proposed for many SOCO instances, the SOCO setting has two limitations. First, the hitting cost $f_t$ is revealed before making action $y_t$, i.e., SOCO requires one step exact prediction of $f_t$. Second, the switching cost in SOCO only depends on one previous action in the form $c(y_t, y_{t-1})$, so only one step of memory is considered. In this paper, our goal is to derive competitive algorithms (as exist for SOCO) in more general settings where more than one step of memory is considered. Working with the general model of OCO with memory is too ambitious for this goal. Instead, we introduce a model of OCO with *structured* memory that generalizes SOCO, and is motivated by a nontrivial connection with online control (as shown in Section 4.2).

Specifically, we consider a loss function $g_t$ at time step $t$ that can be decomposed as the sum of a hitting cost function $f_t : \mathbb{R}^d \to \mathbb{R}^+ \cup \{0\}$ and a switching cost function $c : \mathbb{R}^{d \times (p+1)} \to \mathbb{R}^+ \cup \{0\}$. Additionally, we assume that the switching cost has the form:

$$c(y_{t:t-p}) = \frac{1}{2} \left\| y_t - \sum_{i=1}^{p} C_i y_{t-i} \right\|_2^2,$$

with known $C_i \in \mathbb{R}^{d \times d}, i = 1, \cdots, p$. Note that SOCO is a special case $p = 1$ and $C_1 = I$. As we show in Section 4.2, this form connects online optimization with online control. Intuitively, this connection results from the fact that the hitting cost penalizes the agent for deviating from an optimal point sequence, while the switching cost captures the cost of implementing a control action. Specifically, suppose $y_t$ is a robot's position at $t$, and then the classical SOCO switching cost $\|y_t - y_{t-1}\|_2$ is approximately its velocity. Under our new switching cost, we can represent acceleration by $\|y_t - 2y_{t-1} + y_{t-2}\|_2$, and many other higher-order dynamics.

To summarize, we consider an online agent and an offline adversary interacting as follows in each time step $t$, and we assume $y_i$ is already fixed for $i = -p, -(p-1), \cdots, 0$.

1. The adversary reveals a function $h_t$ and a convex *estimation set* $\Omega_t \subseteq \mathbb{R}^d$. We assume $h_t$ is both $m$-strongly convex and $l$-strongly smooth, and that $\arg\min_y h_t(y) = \mathbf{0}$.
2. The agent picks $y_t \in \mathbb{R}^d$.
3. The adversary picks $v_t \in \Omega_t$.
4. The agent incurs *hitting cost* $f_t(y_t) = h_t(y_t - v_t)$ and *switching cost* $c(y_{t:t-p})$.

Notice that the hitting cost $f_t$ is revealed to the online agent in two separate steps. The geometry of $f_t$ (given by $h_t$ whose minimizer is at $\mathbf{0}$) is revealed before the agent picks $y_t$. After $y_t$ is picked, the minimizer $v_t$ of $f_t$ is revealed.

Unlike SOCO, due to the uncertainty about $v_t$, the agent cannot determine the exact value of the hitting cost it incurs at time step $t$ when determining its action $y_t$. To keep the problem tractable, we

**Algorithm 1:** Regularized OBD (ROBD), Goel et al. [24]

---

**Parameter:** $\lambda_1 \geq 0, \lambda_2 \geq 0$
**for** $t = 1$ **to** $T$ **do**

     **Input:** Hitting cost function $f_t$, previous decision points $y_{t-p}, \cdots, y_{t-1}$
     $v_t \leftarrow \arg\min_y f_t(y)$
     $y_t \leftarrow \arg\min_y f_t(y) + \lambda_1 c(y, y_{t-1:t-p}) + \frac{\lambda_2}{2} \|y - v_t\|_2^2$
     **Output:** $y_t$

---

assume an estimation set $\Omega_t$, which contains all possible $v_t$'s, is revealed to bound the uncertainty. The agent can leverage this information when picking $y_t$. SOCO is a special case where $\Omega_t$ contains only one point, i.e., $\Omega_t = \{v_t\}$, and then the agent has a precise estimate of the minimizer $v_t$ when choosing its action [23, 24]. Like SOCO, the offline optimal cost in the structured memory model is defined as $\texttt{cost}(OPT) = \min_{y_1 \ldots y_T} \sum_{t=1}^{T} f_t(y_t) + c(y_{t:t-p})$ given the full sequence $\{f_t\}_{t=1}^T$.

## 3 Algorithms for OCO with memory

In OCO with structured memory, there is a key differentiation depending on whether the agent has knowledge of the hitting cost function (both $h_t$ and $v_t$) when choosing its action or not, i.e., whether the estimation set $\Omega_t$ is a single point, $v_t$, or not. We deal with each case in turn in the following.

### 3.1 Case 1: exact prediction of $v_t$ ($\Omega_t = \{v_t\}$)

We first study the simplest case where $\Omega_t = \{v_t\}$. Recall that $\Omega_t$ is the convex set which contains all possible $v_t$ and so, in this case, the agent has exact knowledge of the hitting cost when picking action. This assumption, while strict, is standard in the SOCO literature, e.g., [23, 24]. It is appropriate for situations where the cost function can be observed before choosing an action, e.g., [30, 26, 23].

Our main result in this setting is the following theorem, which shows that the ROBD algorithm (Algorithm 1), which is the state-of-the-art algorithm for SOCO, performs well in the more general case of structured memory. Note that, in this setting, the smoothness parameter $l$ of hitting cost functions is not involved in the competitive ratio bound.

**Theorem 1.** *Suppose the hitting cost functions are $m-$strongly convex and the switching cost is given by $c(y_{t:t-p}) = \frac{1}{2} \|y_t - \sum_{i=1}^p C_i y_{t-i}\|_2^2$, where $C_i \in \mathbb{R}^{d \times d}$ and $\sum_{i=1}^p \|C_i\|_2 = \alpha$. The competitive ratio of ROBD with parameters $\lambda_1$ and $\lambda_2$ is upper bounded by:*

$$\max\left\{ \frac{m + \lambda_2}{m\lambda_1}, \frac{\lambda_1 + \lambda_2 + m}{(1 - \alpha^2)\lambda_1 + \lambda_2 + m} \right\},$$

*if $\lambda_1 > 0$ and $(1-\alpha^2)\lambda_1 + \lambda_2 + m > 0$. If $\lambda_1$ and $\lambda_2$ satisfy $m + \lambda_2 = \frac{m + \alpha^2 - 1 + \sqrt{(m + \alpha^2 - 1)^2 + 4m}}{2} \cdot \lambda_1$, then the competitive ratio is:*

$$\frac{1}{2}\left( 1 + \frac{\alpha^2 - 1}{m} + \sqrt{\left(1 + \frac{\alpha^2 - 1}{m}\right)^2 + \frac{4}{m}} \right).$$

The proof of Theorem 1 is given in Appendix C. To get insight into Theorem 1, first consider the case when $\alpha$ is a constant. In this case, the competitive ratio is of order $O(1/m)$, which highlights that the challenging setting is when $m$ is small. It is easy to see that this upper bound is in fact tight. To see this, note that the case of SOCO with $\ell_2$ squared switching cost considered in Goel and Wierman [23], Goel et al. [24] is a special case where $p = 1, C_1 = I, \alpha = 1$. Substituting these parameters into Theorem 1 gives exactly the same upper bound (including constants) as Goel et al. [24], which has been shown to match a lower bound on the achievable cost of any online algorithm, including constant factors. On the other hand, if we instead assume that $m$ is a fixed positive constant. The competitive ratio can be expressed as $1 + O\left(\alpha^2\right)$. Therefore, the competitive ratio gets worse quickly as $\alpha$ increases. This is also the best possible scaling, achievable via any online algorithm, as we show in Appendix D.

Perhaps surprisingly, the memory length $p$ does not appear in the competitive ratio bound, which contradicts the intuition that the online optimization problem should get harder as the memory length increases. However, it is worth noting that $\alpha$ becomes larger as $p$ increases, so the memory length implicitly impacts the competitive ratio. For example, an interesting form of switching cost is

$$c(y_{t:t-p}) = \frac{1}{2} \Big\| \sum_{i=0}^{p} (-1)^i \binom{p}{i} y_{t-i} \Big\|_2^2,$$

which corresponds to the $p^{\text{th}}$ derivative of $y$ and generalizes SOCO ($p = 1$). In this case, we have $\alpha = 2^p - 1$. Hence $\alpha$ grows exponentially in $p$.

### 3.2 Case 2: inexact prediction of $v_t$ ($v_t \in \Omega_t$)

For general $\Omega_t$, ROBD is no longer enough. It needs to be adapted to handle the uncertainty that results from the estimation set $\Omega_t$. Note that this uncertainty set is crucial for many applications, such as online control with adversarial disturbances.

To handle this additional complexity, we propose Optimistic ROBD (Algorithm 2). Optimistic ROBD is based on two key ideas. The first is to ensure that the algorithm tracks the sequence of actions it would have made if given observations of the true cost functions before choosing an action. To formalize this, we define the *accurate sequence* $\{\hat{y}_1, \cdots, \hat{y}_T\}$ to be the choices of ROBD (Algorithm 1) with $\lambda_1 = \lambda$, $\lambda_2 = 0$ when each hitting cost $f_t$ is revealed before picking $\hat{y}_t$. The goal

---

**Algorithm 2:** Optimistic ROBD

**Parameter:** $\lambda \geq 0$
**for** $t = 1$ **to** $T$ **do**
    **Input:** $v_{t-1}, h_t, \Omega_t$
    Initialize a ROBD instance with $\lambda_1 = \lambda, \lambda_2 = 0$
    Recover $f_{t-1}(y) = h_{t-1}(y - v_{t-1})$
    $\hat{y}_{t-1} \leftarrow \text{ROBD}(f_{t-1}, \hat{y}_{t-p-1:t-2})$
    $\tilde{v}_t \leftarrow \arg\min_{v \in \Omega_t} \min_y h_t(y - v) + \lambda c(y, \hat{y}_{t-1:t-p})$
    Estimate $\tilde{f}_t(y) = h_t(y - \tilde{v}_t)$
    $y_t \leftarrow \text{ROBD}(\tilde{f}_t, \hat{y}_{t-p:t-1})$
    **Output:** $y_t$ (the decision at time step $t$)

---

of Optimistic ROBD (Algorithm 2) is to approximate the accurate sequence. In order to track the accurate sequence, the first step is to recover it up to time step $t - 1$ at time step $t$. To do this, after we observe the previous minimizer $v_{t-1}$, we can compute the accurate choice of ROBD as if both $h_{t-1}$ and $v_{t-1}$ are observed before picking $y_{t-1}$. Therefore, Algorithm 2 can compute the *accurate subsequence* $\{\hat{y}_1, \cdots, \hat{y}_{t-1}\}$ at time step $t$. Picking $y_t$ based on the accurate sequence $\{\hat{y}_1, \cdots, \hat{y}_{t-1}\}$ instead of the noisy sequence $\{y_1, \cdots, y_{t-1}\}$ ensures that the actions do not drift too far from the accurate sequence.

The second key idea is to be optimistic by assuming the adversary will give it $v \in \Omega_t$ that minimizes the cost it will experience. Specifically, before $v_t$ is revealed, the algorithm assumes it is the point in $\Omega_t$ which minimizes the weighted sum $h_t(y - v) + \lambda c(y, \hat{y}_{t-1:t-p})$ if ROBD is implemented with parameter $\lambda$ to pick $y$. This ensures that additional cost is never taken unnecessarily, which could be exploited by the adversary. Note that $\min_y h_t(y - v) + \lambda c(y)$ is strongly convex with respect to $v$ (proof in Appendix E), so it is tractable even if $\Omega_t$ is unbounded.

Our main result in this paper (Theorem 2) bounds the competitive ratio of Optimistic ROBD.

**Theorem 2** (Main result). *Suppose the hitting cost functions are both $m-$strongly convex and $l-$strongly smooth and the switching cost is given by $c(y_{t:t-p}) = \frac{1}{2} \left\| y_t - \sum_{i=1}^{p} C_i y_{t-i} \right\|_2^2$, where $C_i \in \mathbb{R}^{d \times d}$ and $\sum_{i=1}^{p} \|C_i\|_2 = \alpha$. For arbitrary $\eta > 0$, the cost of Optimistic ROBD with parameter $\lambda > 0$, is upper bounded by $K_1 \, cost(OPT) + K_2$, where:*

$$K_1 = (1 + \eta) \max \left\{ \frac{1}{\lambda}, \frac{\lambda + m}{(1 - \alpha^2)\lambda + m} \right\}, K_2 = \lambda \Big( \frac{l}{1 + \eta - \lambda} + \frac{4\alpha^2}{\eta} - \frac{m}{\lambda + m} \Big) \sum_{t=1}^{T} \frac{\|v_t - \tilde{v}_t\|^2}{2}.$$

The proof of Theorem 2 is given in Appendix E. This proof is nontrivial and relies on the two key ideas we mentioned before. Although Theorem 2 does not apply to the case $\lambda = 0$, we discuss it separately in Appendix F. Also, note that we can choose $\eta$ to balance $K_1$ and $K_2$ and obtain a competitive ratio, in particular the smallest $\eta$ such that:

$$\lambda \Big( \frac{l}{1 + \eta - \lambda} + \frac{4\alpha^2}{\eta} - \frac{m}{\lambda + m} \Big) \leq 0.$$

Therefore, we have $\eta = O(l + \alpha^2)$ and $K_2 \leq 0$. So the competitive ratio is upper bounded by:

$$O\left((l + \alpha^2) \max\left\{\frac{1}{\lambda}, \frac{\lambda + m}{(1 - \alpha^2)\lambda + m}\right\}\right).$$

However, the reason we present Theorem 2 in terms of $K_1$ and $K_2$ is that, when the diameter of $\Omega_t$ is small, we can pick a small $\eta$ so that the ratio coefficient $K_1$ will be close to the competitive ratio of ROBD when $v_t$ is known before picking $y_t$. This "beyond-the-worst-case" analysis is useful in many applications and we discuss it more in Section 4.3.

## 4 Application to competitive online control

Goel and Wierman [23] show a connection between SOCO and online control in the setting where disturbance is perfectly known at time step $t$ and the control actuation matrix $B$ is invertible, which leads to the only constant-competitive control policy as far as we know. Since the new proposed OCO with structured memory generalizes SOCO, one may expect its connects to more general dynamical systems. In this section, we present a nontrivial reduction from *Input-Disturbed Squared Regulators (IDSRs)* to OCO with structured memory, leading to the first constant-competitive policy in online control with adversarial disturbance.

### 4.1 Control setting

**Input-disturbed systems.** We focus on systems in controllable canonical form defined by:

$$x_{t+1} = Ax_t + B(u_t + w_t), \qquad (1)$$

where $x_t \in \mathbb{R}^n$ is the state, $u_t \in \mathbb{R}^d$ is the control, and $w_t \in \mathbb{R}^d$ is a potentially adversarial disturbance to the system. We further assume that $(A, B)$ is in controllable canonical form (see the right equation), where each $*$ represents a (possibly) non-zero entry, and the rows of $B$ with 1 are the same rows of $A$ with $*$ [36]. It is well-known that any controllable system can be linearly transformed to the canonical form. This system is more restrictive than the general form in linear systems. We call these *Input-Disturbed* systems, since the disturbance $w_t$ is in the control input/action space. There are many corresponding real-world applications that are well-described by Input-Disturbed systems, e.g., external/disturbance force in robotics [38, 39, 18].

**Squared regulator costs.** We consider the following cost model for the controller:

$$c_t(x_t, u_t) = \frac{q_t}{2} \|x_t\|_2^2 + \frac{1}{2} \|u_t\|_2^2, \qquad (2)$$

where $q_t$ is a positive scalar. The sequence $q_{0:T}$ is picked by the adversary and revealed online. The objective of the controller is to minimize the total control cost $\sum_{t=0}^{T} c_t(x_t, u_t)$. We call this cost model the *Squared Regulator* model, which is a restriction of the classical quadratic cost model. This class of costs is general enough to address a fundamental trade-off in optimal control: the trade-off between the state cost and the control effort [27].

**Disturbances.** In the online control literature, a variety of assumptions have been made about the noise $w_t$. In most works, the assumption is that the exact noise $w_t$ is not known before $u_t$ is taken. Many assume $w_t$ is drawn from a certain known distribution, e.g., Agarwal et al. [5]. Others assume $w_t$ is chosen adversarially subject to $\|w_t\|_2$ being upper bounded by a constant $W$, e.g., Agarwal et al. [4]. In a closely related paper, Goel and Wierman [23] connect SOCO with online control under the assumption that $w_t$ can be observed before picking the control action $u_t$. In contrast, in this paper we assume that the exact $w_t$ is not observable before the agent picks $u_t$. Instead, we assume a convex estimation set $W_t$ (not necessarily bounded) that contains all possible $w_t$ is revealed to the online agent to help the agent decide $u_t$. Our assumption is a generalization of Goel and Wierman [23], where $W_t$ is a one-point set, and Agarwal et al. [4], where $W_t$ is a ball of radius $W$ centered at $\mathbf{0}$. Our setting can also naturally model time-Lipschitz noise, where $w_t$ is chosen adversarially subject to $\|w_t - w_{t-1}\|_2 \leq \epsilon$, by picking $W_t$ as a sphere of radius $\epsilon$ centered at $w_{t-1}$, which has many

real-applications such as smooth disturbances in robotics [38, 39]. Moreover, note that our setting is naturally adaptive because of the estimation set $W_t$ (e.g., controller may choose more aggressive action if $W_t$ is small), which is different from the classic $\mathcal{H}_\infty$ control setting [41].

**Competitive ratio.** Our goal is to develop policies with constant (small) competitive ratios. This is a departure from the bulk of the literature [5, 4, 20, 19], which focuses on designing policies that have low regret compared to the optimal linear controller. We show the optimal linear controller can have cost arbitrarily larger than the offline optimal, via an analytic example (Appendix B). We again denote the offline optimal cost, with full knowledge of the sequence $w_{0:T}$, as $\text{cost}(OPT) = \min_{u_{0:T}} \sum_{t=0}^{T} c_t(x_t, u_t)$. For an online algorithm $ALG$, let $\text{cost}(ALG)$ be its cost on the same disturbance sequence $w_{0:T}$. The competitive ratio is then the worst-case ratio of $\text{cost}(ALG)$ and $\text{cost}(OPT)$ over any disturbance sequence, i.e. $\sup_{w_{0:T}} \text{cost}(ALG)/\text{cost}(OPT)$. We show in Section 4.2 an exact correspondence between this $\text{cost}(OPT)$ and the one defined in Section 2.2, so that the competitive ratio guarantees directly translate.

To the best of our knowledge, the only prior work that studies competitive algorithms for online control is Goel and Wierman [23], which considers a very restricted system with invertible $B$ and known $w_t$ at step $t$. A related line of online optimization research studies *dynamic regret*, or *competitive difference*, defined as the difference between online algorithm cost and the offline optimal [31, 29]. For example, Li et al. [31] bound the dynamic regret of online control with time-varying convex costs with no noise. However, results for the dynamic regret depend on the path-length or variation budget, not just system properties. Bounding the competitive ratio is typically more challenging.

## 4.2 A reduction to OCO with structured memory

We now present a reduction from *IDSR*, introduced in Section 4.1, to OCO with structured memory. This reduction allows us to inherit the competitive ratio bounds on Optimistic ROBD for this class of online control problems. Before presenting the reduction, we first introduce some important notations. The indices of non-zero rows in matrix $B$ in (1) are denoted as $\{k_1, \cdots, k_d\} := \mathcal{I}$. We define operator $\psi : \mathbb{R}^n \to \mathbb{R}^d$ as:

$$\psi(x) = \left(x^{(k_1)}, \cdots, x^{(k_d)}\right)^{\mathsf{T}},$$

which extracts the dimensions in $\mathcal{I}$. Moreover, let $p_i = k_i - k_{i-1}$ for $1 \le i \le n$, where $k_0 = 0$. The controllability index of the canonical-form $(A, B)$ is defined as

---

**Algorithm 3:** Reduction to OCO with memory

**Input:** Transition matrix $A$ and control matrix $B$
**Solver:** OCO with structured memory algorithm ALG
**for** $t = 0$ **to** $T - 1$ **do**
  **Observe:** $x_t$, $W_t$, and $q_{t:t+p-1}$
  **if** $t > 0$ **then**
    $w_{t-1} \leftarrow \psi\left(x_t - Ax_{t-1} - Bu_{t-1}\right)$
    $\zeta_{t-1} \leftarrow w_{t-1} + \sum_{i=1}^{p} C_i \zeta_{t-1-i}$
    $v_{t-1} \leftarrow -\zeta_{t-1}$
  Define $h_t(y) = \frac{1}{2} \sum_{i=1}^{d} \left(\sum_{j=1}^{p_i} q_{t+j}\right) \left(y^{(i)}\right)^2$
  Define $\Omega_t = \{-w - \sum_{i=1}^{p} C_i \zeta_{t-i} \mid w \in W_t\}$
  Feed $v_{t-1}, h_t, \Omega_t$ into ALG
  Obtain ALG's output $y_t$
  $u_t \leftarrow y_t - \sum_{i=1}^{p} C_i y_{t-i}$
  **Output:** $u_t$
**Output:** $u_T = 0$

---

$p = \max\{p_1, \cdots, p_d\}$. We assume that the initial state is zero, i.e., $x_0 = \mathbf{0}$. In the reduction, we also need to use matrices $C_i \in \mathbb{R}^{d \times d}, i = 1, \cdots, p$, which regroup the columns of $A(\mathcal{I}, :)$. We define $C_i$ for $i = 1, \cdots, p$ formally by constructing each of its columns. For $j = 1, \cdots, d$, if $i \le p_j$, the $j$ th column of $C_i$ is the $(k_j + 1 - i)$ th column of $A(\mathcal{I}, :)$; otherwise, the $j$ th column of $C_i$ is $\mathbf{0}$. Formally, for $i \in \{1, \cdots, p\}, j \in \{1, \cdots, d\}$, we have:

$$C_i(:, j) = \begin{cases} A(\mathcal{I}, k_j + 1 - i) & \text{if } i \le p_j \\ \mathbf{0} & \text{otherwise.} \end{cases}$$

Based on coefficients $q_{0:T}$, we define:

$$q_{\min} = \min_{0 \le t \le T-1, 1 \le i \le d} \sum_{j=1}^{p_i} q_{t+j}, \quad q_{\max} = \max_{0 \le t \le T-1, 1 \le i \le d} \sum_{j=1}^{p_i} q_{t+j},$$

where we assume $q_t = 0$ for all $t > T$.

**Theorem 3.** *Consider IDSR where the cost function and dynamics are specified by* (2) *and* (1). *We assume the coefficients $q_{t:t+p-1}$ are observable at step $t$. Any instance of IDSR in controllable canonical form can be reduced to an instance of OCO with structured memory by Algorithm 3.*

A proof and an example of Theorem 3 are given in Appendix G. Notably, $cost(OPT)$ and $cost(ALG)$ remain unchanged in the reduction described by Algorithm 3. In fact, Algorithm 3, when instantiated with Optimistic ROBD, provides an efficient algorithm for online control. It only requires $O(p)$ memory to compute the recursive sequences. As stated in Algorithm 3 the recursive computation of $y_t$ and $\zeta_t$ may have numerical issues. However this can be addressed in a straightforward manner when the algorithm is instantiated with Optimistic ROBD (see Appendix H).

## 4.3 Competitive policy

The reduction in Section 4.2 immediately translates the competitive ratio guarantees in Section 3 into competitive policies. As Theorem 2 suggests, we can tune $\eta$ in Optimistic ROBD based on the quality of prediction. As a result, we present two forms of upper bounds for Algorithm 3 in Corollaries 1 and 2. Notably, Corollary 1 gives a tighter bound where good estimations are available, while Corollary 2 gives a bound that does not depend on the quality of the estimations.

In the first case, we assume that a good estimation of $w_t$ is available before picking $u_t$. Specifically, we assume the diameter of set $W_t$ is upper bounded by $\epsilon_t$ at time step $t$, where $\epsilon_t$ is a small positive constant. We derive Corollary 1 by setting $\eta = 1 + \lambda$ in Theorem 2.

**Corollary 1.** *In IDSR, assume that coefficients $q_{t:t+p-1}$ are observable at time step $t$. Let $\alpha = \sum_{i=1}^{q} \|C_i\|_2$, where $C_i, i = 1, \cdots, p$ are defined as in Section 4.2. When the diameter of $W_t$ is upper bounded by $\epsilon_t$ at time step $t$, the total cost incurred by Algorithm 3 (using Optimistic ROBD with parameter $\lambda$) in the online control problem is upper bounded by $K_1 cost(OPT) + K_2$, where:*

$$K_1 = (2 + \lambda) \cdot \max\left\{\frac{1}{\lambda}, \frac{\lambda + q_{\min}}{(1 - \alpha^2)\lambda + q_{\min}}\right\}, K_2 = \lambda\left(\frac{q_{\max}}{2} + \frac{4\alpha^2}{1 + \lambda} - \frac{q_{\min}}{\lambda + q_{\min}}\right) \cdot \sum_{t=0}^{T-1} \frac{1}{2}\epsilon_t^2.$$

The residue term $K_2$ in Corollary 1 becomes negligible when the total estimation error $\sum_{t=0}^{T-1} \epsilon_t^2$ is small, leading to a pure competitive ratio guarantee. Further, if we ignore $K_2$, the coefficient $K_1$ is only constant factor worse than the ratio we obtain when exact prediction of $w_t$ is available.

However, the bound in Corollary 1 can be significantly worse than the case where exact prediction is available when the diameter of $W_t$ is large or unbounded. Hence we introduce a second corollary that does not use any information about $w_t$ when picking $u_t$. Specifically, we assume the diameter of set $W_t$ cannot be bounded, so the upper bound given in Corollary 1 is meaningless. By picking the parameter $\eta$ such that $\lambda\left(\frac{l}{1+\eta-\lambda} + \frac{4\alpha^2}{\eta} - \frac{m}{\lambda+m}\right) \leq 0$ in Theorem 2, we obtain the following result.

**Corollary 2.** *In IDSR, assume that coefficients $q_{t:t+p-1}$ are observable at time step $t$. Let $\alpha = \sum_{i=1}^{q} \|C_i\|_2$, where $C_i, i = 1, \cdots, p$ are defined as in Section 4.2. The competitive ratio of Algorithm 3, using Optimistic ROBD with $\lambda$, is upper bounded by:*

$$O\left((q_{\max} + 4\alpha^2) \max\left\{\frac{1}{\lambda}, \frac{\lambda + q_{\min}}{(1 - \alpha^2)\lambda + q_{\min}}\right\}\right).$$

Compared with Corollary 1, Corollary 2 gives an upper bound that is independent of the size of $W_t$. It is also a pure constant competitive ratio, without any additive term. However, the ratio is worse than the case where exact prediction of $w_t$ is available, especially when $q_{\max}$ or $\alpha$ is large.

**Contrasting no-regret and constant-competitive guarantees.** The predominant benchmark used in previous work on online control via learning is *static regret* relative to the best linear controller in hindsight, i.e., $u_t = -K^*x_t$ [19, 2, 4, 5, 20, 21, 1]. For example, Agarwal et al. [5] achieve logarithmic regret under stochastic noise and strongly convex loss, and Agarwal et al. [4] achieve $O(\sqrt{T})$ regret under adversarial noise and convex loss. However, the cost of the optimal linear controller may be far from the true offline optimal cost. Goel and Hassibi [22] recently show that there is a linear regret between the optimal offline linear policy and the true offline optimal policy in online LQR control. Thus, achieving small regret may still mean having a significantly larger cost than optimal. We illustrate this difference and our algorithm's performance by a 1-d analytic example (Appendix B), and also numerical experiments in higher dimensions (Section 4.4). In particular, we see that the optimal linear controller can be significantly more costly than the offline optimal controller and that Optimistic ROBD can significantly outperform the optimal linear controller.

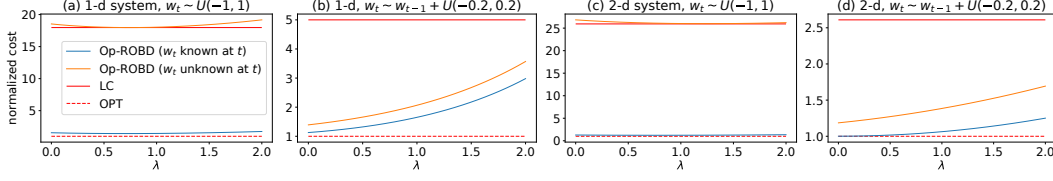

Figure 1: Numerical results of Optimistic ROBD in 1-d and 2-d systems, with different $\lambda$. LC means the best linear controller in hindsight and OPT means the global optimal controller in hindsight. LC is numerically searched in stable linear controller space. We consider two different types of $w_t$: $w_t$ is i.i.d. random/random walk, and also two different settings: $w_t$ is known/unknown at step $t$.

## 4.4 Numerical results

In this section we use simple numerical examples to illustrate the contrast between the best linear controller in hindsight and the optimal offline controller. We also test our algorithm, Optimistic ROBO, and then numerically illustrate that Optimistic ROBD can obtain near-optimal cost and outperform the offline optimal linear controller.

In the first example we consider a simple 1-d system, where the object function is $\sum_{t=0}^{200} 8|x_t|^2 + |u_t|^2$ and the dynamics is $x_{t+1} = 2x_t + u_t + w_t$. For the sequence $\{w_t\}_{t=0}^T$, we consider two cases, in the first case $\{w_t\}_{t=0}^T$ is generated by $w_t \sim \mathcal{U}(-1, 1)$ i.i.d., and in the second case the sequence is generated by $w_{t+1} = w_t + \psi_t$ where $\psi_t \sim \mathcal{U}(-0.2, 0.2)$ i.i.d.. The first case corresponds to unpredictable disturbances, where the estimation set $W_t = (-1, 1)$, and the second to smooth disturbances (i.e., a random walk), where $W_t = w_{t-1} + (-0.2, 0.2)$. For both types of $\{w_t\}_{t=0}^T$, we test Optimistic ROBD algorithms in two settings: $w_t$ is known/unknown at step $t$. In the first setting, $w_t$ is directly given to the algorithm, and in the latter setting, only $W_t$ is given at time step $t$.

The results are shown in Figure 1 (a-b). We see that if $w_t$ is known at step $t$, Optimistic ROBD is much better than the best linear controller in hindsight, and almost matches the true optimal when $w_t$ is smooth. In fact, when $w_t$ is smooth, Optimistic ROBD is much better than the best linear controller even if it does not know $w_t$ at step $t$. Even in the case when $w_t \sim \mathcal{U}(-1, 1)$, and so is extremely unpredictable, Optimistic ROBD's performance still matches the best linear controller, which uses perfect hindsight.

Our second example considers a 2-d system with the following objective and dynamics:

$$\min_{u_t} \sum_{t=0}^{200} 8\|x_t\|_2^2 + \|u_t\|_2^2, \quad \text{s.t. } x_{t+1} = \begin{bmatrix} 0 & 1 \\ -1 & 2 \end{bmatrix} x_t + \begin{bmatrix} 0 \\ 1 \end{bmatrix} u_t + \begin{bmatrix} 0 \\ 1 \end{bmatrix} w_t,$$

where $(A, B)$ is the canonical form of double integrator dynamics. For this 2-d system, similarly, we test the performance of Optimistic ROBD with two types of $w_t$.

The results are shown in Figure 1 (c-d) and reinforce the same observations we observed in the 1-d system. In particular, we see that the optimal linear controller can be significantly more costly than the offline optimal controller and that Optimistic ROBD can outperform the optimal linear controller, sometimes by a significant margin.

## 5 Concluding remarks

We conclude with several open problems and potential future research directions. Our results show the existence of constant-competitive algorithms in a novel class of online optimization with memory, which generalizes SOCO. We also show the existence of constant-competitive control policies in *Input-Disturbed Squared Regulators (IDSRs)*, which is more general than prior work [23]. Following on our work, it will be interesting to understand the breadth of the class of online optimization problems that admit constant-competitive algorithms, and the breath of the class of online control problems where constant-competitive policies exist. Obtaining results (positive or negative) is an important and challenging future direction.

## Broader Impact

Online convex optimization with switching cost (SOCO) has been widely used in commercial and industrial applications such as data centers, power systems, and vehicle charging. By proposing a generalization of SOCO together with new algorithms with competitive ratio guarantees in this setting, this paper opens a new set of applications for online optimization. Additionally, the results provide new fundamental insights about the connection between online optimization and control. However, like many other theoretical contributions, this paper's results are limited to its assumptions, e.g., strongly convex cost functions.

We see no ethical concerns related to the results in this paper.

## Acknowledgments and Disclosure of Funding

This project was supported in part by funding from Raytheon, DARPA PAI, AitF-1637598 and CNS-1518941, with additional support for Guanya Shi provided by the Simoudis Discovery Prize.

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
