[Supplementary Material 1]

# Appendices

## A  Preliminaries

The appendices that follow provide the proofs of the results in the body of the paper. Throughout the proofs we use the following notation to denote the hitting and movement costs of the online learner: $H_t := f_t(y_t)$ and $M_t := c(y_{t:t-p})$, where $y_t$ is the point chosen by the online algorithm at time $t$. Similarly, we denote the hitting and movement costs of the offline optimal as $H_t^* := f_t(y_t^*)$ and $M_t^* := c(y_{t:t-p}^*)$, where $y_t^*$ is the point chosen by the offline optimal at time $t$.

Before moving to the proofs, we summarize a few standard definitions that are used throughout the paper.

**Definition 1.** *A function $f : \mathcal{X} \to \mathbb{R}$ is $m$-strongly convex with respect to a norm $\|\cdot\|$ if for all $x, y$ in the relative interior of the domain of $f$ and $\lambda \in (0, 1)$, we have*

$$f(\lambda x + (1 - \lambda)y) \leq \lambda f(x) + (1 - \lambda)f(y) - \frac{m}{2}\lambda(1 - \lambda)\|x - y\|^2.$$

**Definition 2.** *A function $f : \mathcal{X} \to \mathbb{R}$ is $l$-strongly smooth with respect to a norm $\|\cdot\|$ if $f$ is everywhere differentiable and if for all $x, y$ we have*

$$f(y) \leq f(x) + \langle \nabla f(x), y - x \rangle + \frac{l}{2}\|y - x\|^2.$$

Finally, Lemma 13 in Goel et al. [24] will be useful, and so we restate it here.

**Lemma 1.** *If $f : \mathcal{X} \to \mathbb{R}$ is a $m$-strongly convex function with respect to some norm $\|\cdot\|$, and $v$ is the minimizer of $f$ (i.e. $v = \arg\min_{y \in \mathcal{X}} f(y)$), then we have $\forall y \in \mathcal{X}$,*

$$f(y) \geq f(v) + \frac{m}{2}\|y - v\|^2.$$

## B  Analytic 1-d example

In this section we use simple examples to illustrate the contrast between the best linear controller in hindsight, which is the predominant benchmark, and the optimal offline controller, which is not necessarily linear or static. We highlight analytically that the optimal linear controller can be arbitrarily worse than the optimal offline controller, and then illustrate that analytically that Optimistic ROBD can obtain near-optimal cost.

**Example: a scalar system.** Consider the following scalar system:

$$\min_{u_t} \quad \sum_{t=0}^{T} q|x_t|^2 + |u_t|^2$$

$$\text{s.t.} \quad x_{t+1} = ax_t + u_t + w_t$$

where $a > 1, x_0 = 0$ and $w_t$ is the disturbance. For this system, we have:

$$\frac{\text{cost}(LC)}{\text{cost}(OPT)} > \frac{q + (a-1)^2}{4}, \forall \{w_t\}_{t=0}^{T},$$

where $\text{cost}(LC)$ is the cost of the optimal linear controller in hindsight. Hence, $\text{cost}(LC)/\text{cost}(OPT)$ is arbitrarily large as $q$ and $a$ increase. We emphasize that this lower bound holds for any disturbance sequence, and there exist many sequences making this lower bound even bigger. For example, if $w_t$ is a constant ($w_t = w, \forall t$):

$$\frac{\text{cost}(LC)}{\text{cost}(OPT)} \geq \frac{q + (a-1)^2}{4} \cdot \frac{q + (a-1)^2}{q}.$$

Alternatively, if $w_t = (-1)^t \cdot w$:

$$\frac{\text{cost}(LC)}{\text{cost}(OPT)} \geq \frac{q + (a-1)^2}{4} \cdot \frac{q + (a+1)^2}{q}.$$

Proofs are given in Appendix I. This example highlights that the gap between $\text{cost}(LC)$ and $\text{cost}(OPT)$ can be arbitrarily large for strongly convex costs. Thus, even if an algorithm has no regret compared to the optimal linear controller, it has an unbounded competitive ratio.

Further, we can contrast the competitive ratio of the optimal linear controller derived above with that of Optimistic ROBD. For convenience, assume $\text{cost}(OPT) = T$. First, notice that there exists $\{w_t\}_{t=0}^{T}$ such that $\text{cost}(LC) \geq O(\max\{q, a^4/q\} \cdot T)$ for big enough $a$ and $q$. From Corollary 1, in the case exact prediction of $w_t$ is possible, Optimistic ROBD has $\text{cost}(ALG) \leq O(\max\{1, a^2/q\} \cdot T), \forall \{w_t\}_{t=0}^{T}$, which is orders-of-magnitude lower than $\text{cost}(LC)$.

In the case exact prediction is impossible and the estimation error is $\epsilon_t = w_t - \tilde{w}_t$, Optimistic ROBD guarantees $\text{cost}(ALG) \leq O(\max\{1, a^2/q\} \cdot T + \max\{a^2, q\} \cdot \sum_{t=0}^{T-1} \epsilon_t^2)$ by Corollary 1. Moreover, Corollary 2 gives a constant competitive ratio, $\text{cost}(ALG) \leq O(\max\{q, a^4/q\} \cdot T)$ for any $\{w_t\}_{t=0}^{T}$, which is the same as the lower bound of $\text{cost}(LC)$ we found. Thus, even without any estimate of the noise, our *upper* bound on the cost of Optimistic ROBD matches the *lower* bound on the cost of the optimal linear controller.

## C  Proof of Theorem 1

Our approach is to make use of strong convexity and properties of the hitting cost, the switching cost, and the regularization term to derive an inequality in the form of $H_t + M_t + \Delta\phi_t \leq C(H_t^* + M_t^*)$ for some positive constant $C$, where $\Delta\phi_t$ is the change in potential, which satisfies $\sum_{t=1}^{T} \Delta\phi_t \geq 0$. We will give the formal definition of $\Delta\phi_t$ later. The constant $C$ is then an upper bound for the competitive ratio.

We use $\|\cdot\|$ to denote $\ell_2$ norm or matrix norm induced by $\ell_2$ norm throughout the proof.

By assumption, we have $y_i = y_i^*$ for $i = 0, -1, \cdots, -(p-1)$.

For convenience, we define

$$\phi_t = \frac{\lambda_1 + \lambda_2 + m}{2} \|y_t - y_t^*\|^2.$$

Recall that we define $v_t = \arg\min_y f_t(y)$. Since the function

$$g_t(y) = f_t(y) + \frac{\lambda_1}{2} \left\| y - \sum_{i=1}^{p} C_i y_{t-i} \right\|^2 + \frac{\lambda_2}{2} \|y - v_t\|^2$$

is $(m + \lambda_1 + \lambda_2)-$strongly convex and ROBD selects $y_t = \arg\min_y g_t(y)$, we see that

$$g_t(y_t) + \frac{m + \lambda_1 + \lambda_2}{2} \|y_t - y_t^*\|^2 \le g_t(y_t^*),$$

which implies

$$
\begin{aligned}
H_t &+ \lambda_1 M_t + \left( \phi_t - \sum_{i=1}^{p} \frac{\|C_i\|}{\alpha} \phi_{t-i} \right) \\
&\le \left( H_t^* + \frac{\lambda_2}{2} \|y_t^* - v_t\|^2 \right) + \left( \frac{\lambda_1}{2} \left\| y_t^* - \sum_{i=1}^{p} C_i y_{t-i} \right\|^2 - \sum_{i=1}^{p} \frac{\|C_i\|}{\alpha} \phi_{t-i} \right).
\end{aligned}
\tag{3}
$$

In the following steps, we bound the second term in the right-hand side of (3) by the switching cost of the offline optimal.

$$
\begin{aligned}
\sum_{i=1}^{p} &\frac{\|C_i\|}{\alpha} \phi_{t-i} \\
&= \frac{\lambda_1 + \lambda_2 + m}{2\alpha} \sum_{i=1}^{p} \|C_i\| \cdot \left\| y_{t-i} - y_{t-i}^* \right\|^2 \\
&\ge \frac{\lambda_1 + \lambda_2 + m}{2\alpha^2} \left( \sum_{i=1}^{p} \|C_i\| \cdot \left\| y_{t-i} - y_{t-i}^* \right\| \right)^2 \tag{4a} \\
&\ge \frac{\lambda_1 + \lambda_2 + m}{2\alpha^2} \left( \sum_{i=1}^{p} \left\| C_i y_{t-i} - C_i y_{t-i}^* \right\| \right)^2 \tag{4b} \\
&\ge \frac{\lambda_1 + \lambda_2 + m}{2\alpha^2} \left\| \sum_{i=1}^{p} C_i y_{t-i} - \sum_{i=1}^{p} C_i y_{t-i}^* \right\|^2, \tag{4c}
\end{aligned}
$$

where we use Jensen's Inequality in (4a); the definition of the matrix norm in (4b); the triangle inequality in (4c).

For notation convenience, we define

$$\delta_t = \sum_{i=1}^{p} C_i y_{t-i} - \sum_{i=1}^{p} C_i y_{t-i}^*.$$

Therefore, we obtain that

$$
\frac{\lambda_1}{2}\left\|y_t^* - \sum_{i=1}^{p} C_i y_{t-i}\right\|^2 - \sum_{i=1}^{p}\frac{\|C_i\|}{\alpha}\phi_{t-i}
$$

$$
\leq \frac{\lambda_1}{2}\left\|y_t^* - \sum_{i=1}^{p} C_i y_{t-i}\right\|^2 - \frac{\lambda_1 + \lambda_2 + m}{2\alpha^2}\cdot\|\delta_t\|^2 \tag{5a}
$$

$$
= \frac{\lambda_1}{2}\left\|\left(y_t^* - \sum_{i=1}^{p} C_i y_{t-i}^*\right) - \delta_t\right\|^2 - \frac{\lambda_1 + \lambda_2 + m}{2\alpha^2}\cdot\|\delta_t\|^2
$$

$$
\leq \frac{\lambda_1}{2}\left\|y_t^* - \sum_{i=1}^{p} C_i y_{t-i}^*\right\|^2 + \lambda_1\left\|y_t^* - \sum_{i=1}^{p} C_i y_{t-i}^*\right\|\cdot\|\delta_t\|
$$

$$
+ \frac{\lambda_1}{2}\|\delta_t\|^2 - \frac{\lambda_1 + \lambda_2 + m}{2\alpha^2}\|\delta_t\|^2 \tag{5b}
$$

$$
= \frac{\lambda_1}{2}\left\|y_t^* - \sum_{i=1}^{p} C_i y_{t-i}^*\right\|^2 + \lambda_1\left\|y_t^* - \sum_{i=1}^{p} C_i y_{t-i}^*\right\|\cdot\|\delta_t\|
$$

$$
- \frac{(1-\alpha^2)\lambda_1 + \lambda_2 + m}{2\alpha^2}\|\delta_t\|^2
$$

$$
\leq \frac{\lambda_1}{2}\left\|y_t^* - \sum_{i=1}^{p} C_i y_{t-i}^*\right\|^2 + \frac{\alpha^2\lambda_1^2}{2\left((1-\alpha^2)\lambda_1 + \lambda_2 + m\right)}\left\|y_t^* - \sum_{i=1}^{p} C_i y_{t-i}^*\right\|^2
$$

$$
+ \frac{(1-\alpha^2)\lambda_1 + \lambda_2 + m}{2\alpha^2}\|\delta_t\|^2 - \frac{(1-\alpha^2)\lambda_1 + \lambda_2 + m}{2\alpha^2}\|\delta_t\|^2 \tag{5c}
$$

$$
= \frac{\lambda_1(\lambda_1 + \lambda_2 + m)}{(1-\alpha^2)\lambda_1 + \lambda_2 + m}M_t^*,
$$

where we use (4) in (5a); the triangle inequality in (5b); the AM-GM inequality in (5c).

We also notice that since $f_t$ is $m$-strongly convex, the first term in the right-hand side of (3) can be bounded by

$$
H_t^* + \frac{\lambda_2}{2}\|y_t^* - v_t\|^2 \leq \frac{m + \lambda_2}{m}H_t^*. \tag{6}
$$

Substituting (5) and (6) into (3), we obtain that

$$
H_t + \lambda_1 M_t + \phi_t - \sum_{t=1}^{q}\frac{\|C_i\|}{\alpha}\phi_{t-i}
$$

$$
\leq \frac{m + \lambda_2}{m}H_t^* + \frac{\lambda_1(\lambda_1 + \lambda_2 + m)}{(1-\alpha^2)\lambda_1 + \lambda_2 + m}M_t^*. \tag{7}
$$

Define $\Delta\phi_t = \phi_t - \sum_{t=1}^{q}\frac{\|C_i\|}{\alpha}\phi_{t-i}$. We see that

$$
\sum_{t=1}^{T}\Delta\phi_t = \frac{1}{\alpha}\sum_{i=0}^{q-1}\left(\sum_{j=i+1}^{q}\|C_j\|\right)\phi_{T-i} - \frac{1}{\alpha}\sum_{i=0}^{q-1}\left(\sum_{j=i+1}^{q}\|C_j\|\right)\phi_{-i}.
$$

Since $\phi_t \geq 0, \forall t$ and $\phi_0 = \phi_{-1} = \cdots = \phi_{-q+1} = 0$, we have

$$
\sum_{t=1}^{T}\Delta\phi_t \geq 0. \tag{8}
$$

Summing (7) over timesteps $t = 1, 2, \cdots, T$, we see that

$$
\sum_{t=1}^{T}(H_t + \lambda_1 M_t) + \sum_{t=1}^{T}\Delta\phi_t \leq \sum_{t=1}^{T}\left(\frac{m + \lambda_2}{m}H_t^* + \frac{\lambda_1(\lambda_1 + \lambda_2 + m)}{(1-\alpha^2)\lambda_1 + \lambda_2 + m}M_t^*\right).
$$

Using (8), we obtain that

$$\sum_{t=1}^{T}(H_t + \lambda_1 M_t) \le \sum_{t=1}^{T}\left(\frac{m+\lambda_2}{m}H_t^* + \frac{\lambda_1(\lambda_1+\lambda_2+m)}{(1-\alpha^2)\lambda_1+\lambda_2+m}M_t^*\right), \tag{9}$$

which implies

$$\sum_{t=1}^{T}(H_t + M_t) \le \sum_{t=1}^{T}\left(\frac{m+\lambda_2}{m\lambda_1}H_t^* + \frac{\lambda_1+\lambda_2+m}{(1-\alpha^2)\lambda_1+\lambda_2+m}M_t^*\right).$$

## D  Lower bound of online optimization with structured memory

Theorem 1 considers the problem setting where the hitting cost functions are $m-$strongly convex in feasible set $\mathcal{X}$ and the switching cost is given by $c(y_{t:t-p}) = \frac{1}{2}\left\|y_t - \sum_{i=1}^{p}C_i y_{t-i}\right\|_2^2$, where $C_i \in \mathbb{R}^{d\times d}$ and $\sum_{i=1}^{p}\|C_i\|_2 = \alpha$. We prove that the competitive ratio provided in Theorem 1 is optimal in parameters $\alpha$ and $m$ by showing a lower bound for a specific sequence of hitting costs and a specific form of switching cost, $c(y_t, y_{t-1}) = \frac{1}{2}\|y_t - \alpha y_{t-1}\|_2^2$.

Notice that making improvements on the competitive ratio is still possible if we consider more specific matrix $C_i$ or adding more assumptions on the hitting cost functions.

**Theorem 4.** *When the hitting cost functions are $m-$strongly convex in feasible set $\mathcal{X}$ and the switching cost is given by $c(y_t, y_{t-1}) = \frac{1}{2}\|y_t - \alpha y_{t-1}\|_2^2$ for a constant $\alpha \ge 1$, the competitive ratio of any online algorithm is lower bounded by*

$$\frac{1}{2}\left(1 + \frac{\alpha^2-1}{m} + \sqrt{\left(1+\frac{\alpha^2-1}{m}\right)^2 + \frac{4}{m}}\right).$$

Theorem 4 is a generalization of [24][Theorem 1], which only considers the case when $\alpha = 1$. Our proof uses a parallel approach but extends it to general $\alpha$. Before giving the proof of Theorem 4, we first prove the generalization of [24][Lemma 7]. To simplify presentation in the proofs, we use $\mathcal{K}(n,y)$ to denote the set $\{y \in \mathbb{R}^{n+2} \mid y_i \in \mathbb{R}, y_0 = 0, y_{n+1} = y\}$.

**Lemma 2.** *For $m > 0$ and $\alpha \ge 1$, define*

$$a_n = 2\min_{y^* \in \mathcal{K}(n,1)}\left(\sum_{i=1}^{n}\frac{m}{2}(y_i^*)^2 + \sum_{i=1}^{n+1}\frac{1}{2}(y_i^* - \alpha y_{i-1}^*)^2\right).$$

*Then we have $\lim_{n\to\infty} a_n = \frac{-m-\alpha^2+1+\sqrt{(m+\alpha^2-1)^2+4m}}{2}$.*

*Proof of Lemma 2.* Using a parallel approach to [24][Lemma 7], we can show that sequence $\{a_n\}$ satisfies the recursive relationship

$$a_{n+1} = \frac{a_n + m}{a_n + m + \alpha^2}.$$

Solving the equation $y = \frac{y+m}{y+m+\alpha^2}$, we find the two fixed points of the recursive relationship $a_{n+1} = \frac{a_n+m}{a_n+m+\alpha^2}$ are

$$y_1 = \frac{-m-\alpha^2+1+\sqrt{(m+\alpha^2-1)^2+4m}}{2},$$

and

$$y_2 = \frac{-m-\alpha^2+1-\sqrt{(m+\alpha^2-1)^2+4m}}{2}.$$

Notice that for $i = 1, 2$, we have

$$m - (m+\alpha^2)y_i = -(1-y_i)y_i.$$

Using this property, we obtain

$$a_{n+1} - y_1 = \frac{a_n + m}{a_n + m + \alpha^2} - y_1 = \frac{(1-y_1)a_n + m - (m+\alpha^2)y_1}{a_n + m + \alpha^2} = \frac{(1-y_1)(a_n - y_1)}{a_n + m + \alpha^2}, \quad (10)$$

and

$$a_{n+1} - y_2 = \frac{a_n + m}{a_n + m + \alpha^2} - y_2 = \frac{(1-y_2)a_n + m - (m+\alpha^2)y_2}{a_n + m + \alpha^2} = \frac{(1-y_2)(a_n - y_2)}{a_n + m + \alpha^2}. \quad (11)$$

Notice that $a_{n+1} - y_2 > 0$. By dividing equations (10) and (11), we obtain

$$\left( \frac{a_{n+1} - y_1}{a_{n+1} - y_2} \right) = \frac{1 - y_1}{1 - y_2} \cdot \left( \frac{a_n - y_1}{a_n - y_2} \right), \forall n \geq 0.$$

Solving this in a parallel way to [24][Lemma 7], we get

$$a_n = \left( 1 - \left( \frac{1-y_1}{1-y_2} \right)^{n+1} \right)^{-1} \left( y_1 - y_2 \cdot \left( \frac{1-y_1}{1-y_2} \right)^{n+1} \right).$$

Since $0 < \left( \frac{1-y_1}{1-y_2} \right) < 1$, we have

$$\lim_{n \to \infty} a_n = y_1 = \frac{-m - \alpha^2 + 1 + \sqrt{(m+\alpha^2-1)^2 + 4m}}{2}. \quad (12)$$

$\square$

Now we come back to the proof of Theorem 4.

*Proof of Theorem 4.* We consider the counterexample where the starting point of the algorithm and the offline adversary is $y_0 = y_0^* = 0$, and the hitting cost functions are

$$f_t(y) = \begin{cases} \frac{m}{2} y^2 & t \in \{1, 2, \cdots, n\} \\ \frac{m'}{2} (y-1)^2 & t = n+1 \end{cases}$$

for some large parameter $m'$ that we choose later.

By a parallel approach to [24][Theorem 1], we can show the cost incurred by any online algorithm has the lower bound

$$\text{cost}(ALG) \geq \min_y \left( \frac{1}{2} y^2 + \frac{m'}{2} (y-1)^2 \right) = \frac{1}{2 \left( 1 + \frac{1}{m'} \right)}. \quad (13)$$

In contrast to the case when $\alpha = 1$, the offline adversary can leverage the factor $\alpha$ to approach 1 quicker if $\alpha > 1$.

Let the sequence of points the adversary chooses be $y^* = (y_0^*, y_1^*, \cdots, y_{n+1}^*) \in \mathbb{R}^{n+2}$. We compute the cost incurred by the adversary as follows.

$$a_n = 2 \min_{y^* \in \mathcal{K}(n,1)} \sum_{i=1}^{n+1} (H_i^* + M_i^*)$$

$$= 2 \min_{y^* \in \mathcal{K}(n,1)} \left( \sum_{i=1}^{n} \frac{m}{2} (y_i^*)^2 + \sum_{i=1}^{n+1} \frac{1}{2} (y_i^* - \alpha y_{i-1}^*)^2 \right).$$

In words, $a_n$ is twice the minimal offline cost subject to the constraints $y_0^* = 0, y_{n+1}^* = 1$. Recall that we have derived the limiting behavior of the offline costs as $n \to \infty$ for general $\alpha$ in the Lemma 2. Given Lemma 2, the total cost of the offline adversary will be $\frac{a_n}{2}$. Finally, applying (13), we know $\forall n$ and $\forall m' > 0$,

$$\frac{\text{cost}(ALG)}{\text{cost}(ADV)} \geq \frac{\frac{1}{2(1+\frac{1}{m'})}}{\frac{a_n}{2}} = \frac{1}{(1+\frac{1}{m'})a_n}.$$

By taking the limit $n \to \infty$ and $m' \to \infty$ and using Lemma 2, we obtain

$$\frac{\text{cost}(ALG)}{\text{cost}(OPT)} = \lim_{n,m' \to \infty} \frac{\text{cost}(ALG)}{\text{cost}(ADV)} \geq \frac{1}{2} \left( 1 + \frac{\alpha^2 - 1}{m} + \sqrt{\left( 1 + \frac{\alpha^2 - 1}{m} \right)^2 + \frac{4}{m}} \right).$$

$\square$

# E Proof of Theorem 2

We use $\|\cdot\|$ to denote $\ell_2$ norm or matrix norm induced by $\ell_2$ norm throughout the proof. Before giving the proof of Theorem 2, we first prove three lemmas that we use later.

Recall that ROBD with parameters $\lambda_1 = \lambda, \lambda_2 = 0$ minimizes a weighted sum of the *hitting cost* $f_t$ and the *switching cost* $c$. To pick the appropriate estimation of $v_t$ from the set $\Omega_t$, we want to study when the previous decision points $\hat{y}_{t-p:t-1}$ is fixed, how the position of $v_t$ will affect the minimum of this weighted sum. By a change of variable, we see this is equivalent to study when the hitting cost function is fixed, how the sum $\sum_{i=1}^{p} C_i \hat{y}_{t-i}$ will affect the weighted sum. We use $x$ to denote the sum $\sum_{i=1}^{p} C_i \hat{y}_{t-i}$ in Lemma 3.

**Lemma 3.** *Suppose function $f : \mathbb{R}^d \to \mathbb{R}$ is $m$-strongly convex. Define function $g : \mathbb{R}^d \to \mathbb{R}$ as*

$$g(x) = \min_y f(y) + \frac{\lambda}{2} \|y - x\|^2.$$

*Then $g$ is $\frac{\lambda m}{\lambda + m}$-strongly convex.*

*Proof of Lemma 3.* Due to the definition of strongly convexity, we only need to show that for all $x_1, x_2 \in \mathbb{R}^d$ and $\eta \in (0, 1)$, we have

$$g(\eta x_1 + (1-\eta)x_2) \leq \eta g(x_1) + (1-\eta)g(x_2) - \frac{\lambda m}{2(\lambda + m)} \eta(1-\eta) \|x_1 - x_2\|^2.$$

For convenience, we define

$$y_1 := \arg\min_y f(y) + \frac{\lambda}{2} \|y - x_1\|^2,$$

and

$$y_2 := \arg\min_y f(y) + \frac{\lambda}{2} \|y - x_2\|^2.$$

We have that

$$\eta g(x_1) + (1-\eta)g(x_2) - \frac{\lambda m}{2(\lambda + m)} \eta(1-\eta) \|x_1 - x_2\|^2$$

$$= \eta f(y_1) + (1-\eta)f(y_2) + \frac{\eta\lambda}{2} \|y_1 - x_1\|^2 + \frac{(1-\eta)\lambda}{2} \|y_2 - x_2\|^2 - \frac{\lambda m}{2(\lambda + m)} \eta(1-\eta) \|x_1 - x_2\|^2 \tag{14a}$$

$$\geq f(\eta y_1 + (1-\eta)y_2) + \frac{m}{2}\eta(1-\eta) \|y_1 - y_2\|^2 - \frac{\lambda m}{2(\lambda + m)} \eta(1-\eta) \|x_1 - x_2\|^2$$

$$+ \frac{\eta\lambda}{2} \|y_1 - x_1\|^2 + \frac{(1-\eta)\lambda}{2} \|y_2 - x_2\|^2 \tag{14b}$$

$$\geq g(\eta x_1 + (1-\eta)x_2) + \frac{m}{2}\eta(1-\eta) \|y_1 - y_2\|^2 - \frac{\lambda m}{2(\lambda + m)} \eta(1-\eta) \|x_1 - x_2\|^2$$

$$+ \frac{\eta\lambda}{2} \|y_1 - x_1\|^2 + \frac{(1-\eta)\lambda}{2} \|y_2 - x_2\|^2 - \frac{\lambda}{2} \|\eta(y_1 - x_1) + (1-\eta)(y_2 - x_2)\|^2 \tag{14c}$$

$$\geq g(\eta x_1 + (1-\eta)x_2) + \frac{m}{2}\eta(1-\eta) \|y_1 - y_2\|^2 - \frac{\lambda m}{2(\lambda + m)} \eta(1-\eta) \|x_1 - x_2\|^2$$

$$+ \frac{\eta(1-\eta)\lambda}{2} \|(y_1 - y_2) - (x_1 - x_2)\|^2,$$

where in (14a) and (14c) we use the definition of function $g$; in (14b) we use the fact that $f$ is $m-$strongly convex; in (14c) we use function $\frac{\lambda}{2} \|\cdot\|^2$ is $\lambda-$strongly convex.

Notice that

$$m \|y_1 - y_2\|^2 - \frac{\lambda m}{\lambda + m} \|x_1 - x_2\|^2 + \lambda \|(y_1 - y_2) - (x_1 - x_2)\|^2$$

$$\geq m \|y_1 - y_2\|^2 - \frac{\lambda m}{\lambda + m} \|x_1 - x_2\|^2 + \lambda \|y_1 - y_2\|^2 + \lambda \|x_1 - x_2\|^2 - 2\lambda \|y_1 - y_2\| \cdot \|x_1 - x_2\|$$

$$= (m + \lambda) \|y_1 - y_2\|^2 + \frac{\lambda^2}{m + \lambda} \|x_1 - x_2\|^2 - 2\lambda \|y_1 - y_2\| \cdot \|x_1 - x_2\|$$

$$\geq 0.$$

$$(15)$$

Substituting (15) into (14) finishes the proof. □

In the second lemma, we show that if a function $f$ is strongly smooth, the function value $f(y)$ at point $y$ can be upper bounded by a weighted sum of the function value $f(x)$ at another point $x$ and the squared distance between $x$ and $y$.

**Lemma 4.** *If $f : \mathbb{R}^d \to \mathbb{R}^+ \cup \{0\}$ is convex and $l$-strongly smooth, we have for all $x, y \in \mathbb{R}^d$, the inequality*

$$f(y) \leq (1 + \eta) f(x) + \left(1 + \frac{1}{\eta}\right) \cdot \frac{l}{2} \|x - y\|^2$$

*holds for all $\eta > 0$.*

*Proof of Lemma 4.* Let $v := \arg \min_z f(z)$.

Using the property of $l$-strongly smoothness, we see that

$$f(x) \geq f(v) + \langle \nabla f(v), x - v \rangle + \frac{1}{2l} \|\nabla f(x) - \nabla f(v)\|^2 \tag{16a}$$

$$\geq \frac{1}{2l} \|\nabla f(x)\|^2, \tag{16b}$$

where we use [13][Lemma 3.5] in (16a); we use $f(v) \geq 0, \nabla f(v) = 0$ in (16b).

Therefore, we obtain that

$$f(y) \leq f(x) + \langle \nabla f(x), y - x \rangle + \frac{l}{2} \|y - x\|^2 \tag{17a}$$

$$\leq f(x) + \|\nabla f(x)\| \cdot \|y - x\| + \frac{l}{2} \|y - x\|^2 \tag{17b}$$

$$\leq f(x) + \frac{\eta}{2l} \|\nabla f(x)\|^2 + \frac{l}{2\eta} \|y - x\|^2 + \frac{l}{2} \|y - x\|^2 \tag{17c}$$

$$\leq f(x) + \eta f(x) + \left(1 + \frac{1}{\eta}\right) \cdot \frac{l}{2} \|y - x\|^2 \tag{17d}$$

$$= (1 + \eta) f(x) + \left(1 + \frac{1}{\eta}\right) \cdot \frac{l}{2} \|y - x\|^2,$$

where we use that $f$ is $l$-strongly smooth in (17a); Cauchy-Schwarz Inequality in (17b); AM-GM inequality in (17c); (16) in (17d). □

Recall that $\hat{y}_t$ is the decision point of ROBD which knows tha exact $v_t$ before picking $\hat{y}_t$. $y_t$ is the decision point of Optimistic ROBD which cannot observe the exact $v_t$ before picking $y_t$. In the third lemma, we show that $y_t$ and $\hat{y}_t$ will be close to each other once the estimated minimizer $\tilde{v}_t$ computed by Optimistic ROBD is close to the true minimizer $v_t$.

**Lemma 5.** *Under the same assumptions as Theorem 2, the distance between $y_t$ and $\hat{y}_t$ can be upper bounded by*

$$\|y_t - \hat{y}_t\| \leq 2 \|\zeta_t\|,$$

*where $\zeta_t = v_t - \tilde{v}_t$.*

*Proof of Lemma 5.* Recall that by definition, the real hitting cost function which we used to pick $\hat{y}_t$ is $f_t(y) = h_t(y - v_t)$, and the estimated hitting cost function which we used to pick $y_t$ is given by $\tilde{f}_t(y) = h_t(y - \tilde{v}_t)$. Therefore, we have $\tilde{f}_t(y) = f_t(y + \zeta_t)$.

Since $\hat{y}_t = ROBD(f_t, \hat{y}_{t-1:t-q}) = \arg\min_y f_t(y) + \lambda c(y, \hat{y}_{t-1:t-p})$, by strongly convexity, we have that

$$
\begin{aligned}
&f_t(\hat{y}_t) + \frac{\lambda}{2} \left\| \hat{y}_t - \sum_{i=1}^{p} C_i \hat{y}_{t-i} \right\|^2 + \frac{m+\lambda}{2} \left\| \hat{y}_t - y_t - \zeta_t \right\|^2 \\
&\leq f_t(y_t + \zeta_t) + \frac{\lambda}{2} \left\| y_t + \zeta_t - \sum_{i=1}^{p} C_i \hat{y}_{t-i} \right\|^2.
\end{aligned}
\tag{18}
$$

Similarly, using $y_t = ROBD(\tilde{f}_t, \hat{y}_{t-1:t-q}) = \arg\min_y f_t(y + \zeta_t) + \lambda c(y, \hat{y}_{t-1:t-p})$, we obtain that

$$
\begin{aligned}
&f_t(y_t + \zeta_t) + \frac{\lambda}{2} \left\| y_t - \sum_{i=1}^{p} C_i \hat{y}_{t-i} \right\|^2 + \frac{m+\lambda}{2} \left\| \hat{y}_t - y_t - \zeta_t \right\|^2 \\
&\leq f_t(\hat{y}_t) + \frac{\lambda}{2} \left\| \hat{y}_t - \zeta_t - \sum_{i=1}^{p} C_i \hat{y}_{t-i} \right\|^2.
\end{aligned}
\tag{19}
$$

Adding (18) and (19) together, we obtain that

$$
\begin{aligned}
&(m+\lambda) \left\| \hat{y}_t - y_t - \zeta_t \right\|^2 \\
&\leq \frac{\lambda}{2} \left( \left\| y_t + \zeta_t - \sum_{i=1}^{p} C_i \hat{y}_{t-i} \right\|^2 - \left\| y_t - \sum_{i=1}^{p} C_i \hat{y}_{t-i} \right\|^2 + \left\| \hat{y}_t - \zeta_t - \sum_{i=1}^{p} C_i \hat{y}_{t-i} \right\|^2 - \left\| \hat{y}_t - \sum_{i=1}^{p} C_i \hat{y}_{t-i} \right\|^2 \right) \\
&= \lambda \zeta_t^{\mathsf{T}} (y_t + \zeta_t - \hat{y}_t) \\
&\leq \lambda \left\| \zeta_t \right\| \cdot \left\| \hat{y}_t - y_t - \zeta_t \right\|.
\end{aligned}
\tag{20}
$$

Therefore, we see that

$$
\left\| \hat{y}_t - y_t - \zeta_t \right\| \leq \left\| \zeta_t \right\|,
$$

which implies

$$
\left\| y_t - \hat{y}_t \right\| \leq 2 \left\| \zeta_t \right\|.
$$

$\square$

Now we come back to the proof of Theorem 2.

Define function $\psi : \mathbb{R}^d \to \mathbb{R}^+ \cup \{0\}$ as

$$
\psi(v) = \min_y h_t(y - v) + \lambda c(y, \hat{y}_{t-1:t-q}).
$$

By a change of variable $y \leftarrow z + v$, we can rewrite function $\psi$ as

$$
\psi(v) = \min_z h_t(z) + \frac{\lambda}{2} \left\| z - \left( -v + \sum_{i=1}^{p} C_i \hat{y}_{t-i} \right) \right\|^2.
\tag{21}
$$

By Lemma 3, we see that function $\psi$ is $\frac{\lambda m}{\lambda + m}$-strongly convex.

Recall that

$$
y_t = ROBD(\tilde{f}_t, \hat{y}_{t-1:t-q}) = \arg\min_y h_t(y - \tilde{v}_t) + \lambda c(y, \hat{y}_{t-1:t-q}),
\tag{22}
$$

and

$$
\hat{y}_t = ROBD(f_t, \hat{y}_{t-1:t-q}) = \arg\min_y h_t(y - v_t) + \lambda c(y, \hat{y}_{t-1:t-q}).
\tag{23}
$$

Since $\tilde{v}_t$ minimizes $\psi$ and $\psi$ is $\frac{\lambda m}{\lambda + m}$-strongly convex, using (22) and (23), we obtain that

$$
\begin{aligned}
& h_t(y_t - \tilde{v}_t) + \frac{\lambda}{2} \left\| y_t - \sum_{i=1}^{p} C_i \hat{y}_{t-i} \right\|^2 + \frac{1}{2} \cdot \frac{m\lambda}{\lambda + m} \|v_t - \tilde{v}_t\|^2 \\
& \leq h_t(\hat{y}_t - v_t) + \frac{\lambda}{2} \left\| \hat{y}_t - \sum_{i=1}^{p} C_i \hat{y}_{t-i} \right\|^2 .
\end{aligned}
\tag{24}
$$

Using Lemma 4, we see that for any $\eta_1 > 0$,

$$
\frac{1}{1+\eta_1} h_t(y_t - v_t) \leq h_t(y_t - \tilde{v}_t) + \frac{l}{2\eta_1} \|v_t - \tilde{v}_t\|^2 .
\tag{25}
$$

Since function $\frac{\lambda}{2} \|y_t - y\|^2$ is $\lambda$-strongly smooth in $y$, by Lemma 4, we see that for any $\eta_2 > 0$,

$$
\frac{1}{1+\eta_2} \cdot \frac{\lambda}{2} \left\| y_t - \sum_{i=1}^{p} C_i y_{t-i} \right\|^2 \leq \frac{\lambda}{2} \left\| y_t - \sum_{i=1}^{p} C_i \hat{y}_{t-i} \right\|^2 + \frac{\lambda}{2\eta_2} \left\| \sum_{i=1}^{p} C_i (y_{t-i} - \hat{y}_{t-i}) \right\|^2 . \tag{26}
$$

Notice that

$$
\frac{1}{2} \left\| \sum_{i=1}^{p} C_i (y_{t-i} - \hat{y}_{t-i}) \right\|^2 \leq \frac{1}{2} \left( \sum_{i=1}^{p} \|C_i\| \cdot \|y_{t-i} - \hat{y}_{t-i}\| \right)^2 \tag{27a}
$$

$$
\leq \frac{\alpha}{2} \left( \sum_{i=1}^{p} \|C_i\| \cdot \|y_{t-i} - \hat{y}_{t-i}\|^2 \right) \tag{27b}
$$

$$
\leq 2\alpha \left( \sum_{i=1}^{p} \|C_i\| \cdot \|\tilde{v}_{t-i} - v_{t-i}\|^2 \right), \tag{27c}
$$

where we use the triangle inequality and the definition of matrix norm in (27a); Jensen's inequality in (27b); Lemma 5 in (27c).

Substituting (27) into (26) gives

$$
\frac{1}{1+\eta_2} \cdot \frac{\lambda}{2} \left\| y_t - \sum_{i=1}^{p} C_i y_{t-i} \right\|^2 \leq \frac{\lambda}{2} \left\| y_t - \sum_{i=1}^{p} C_i \hat{y}_{t-i} \right\|^2 + \frac{2\alpha\lambda}{\eta_2} \left( \sum_{i=1}^{p} \|C_i\| \cdot \|\tilde{v}_{t-i} - v_{t-i}\|^2 \right) .
\tag{28}
$$

Substituting (25) and (28) into (24), we obtain that

$$
\begin{aligned}
& \frac{1}{1+\eta_1} h_t(y_t - v_t) + \frac{\lambda}{2(1+\eta_2)} \left\| y_t - \sum_{i=1}^{p} C_i y_{t-i} \right\|^2 \\
& \leq h_t(\hat{y}_t - v_t) + \frac{\lambda}{2} \left\| \hat{y}_t - \sum_{i=1}^{p} C_i \hat{y}_{t-i} \right\|^2 + \left( \frac{l}{\eta_1} - \frac{m\lambda}{\lambda + m} \right) \cdot \frac{1}{2} \|v_t - \tilde{v}_t\|^2 + \frac{2\alpha\lambda}{\eta_2} \left( \sum_{i=1}^{p} \|C_i\| \cdot \|\tilde{v}_{t-i} - v_{t-i}\|^2 \right) .
\end{aligned}
\tag{29}
$$

Summing up (29) over all time steps, we see that

$$
\begin{aligned}
& \min\{ \frac{1}{1+\eta_1}, \frac{\lambda}{1+\eta_2} \} \sum_{t=1}^{T} (H_t + M_t) \\
& \leq \sum_{t=1}^{T} \left( \hat{H}_t + \lambda \hat{M}_t \right) + \left( \frac{l}{\eta_1} + \frac{4\alpha^2 \lambda}{\eta_2} - \frac{m\lambda}{\lambda + m} \right) \cdot \sum_{t=1}^{T} \frac{1}{2} \|v_t - \tilde{v}_t\|^2 .
\end{aligned}
\tag{30}
$$

We pick $\eta_2 = \eta$ and $\eta_1 = \frac{1+\eta-\lambda}{\lambda}$ so that $\frac{1}{1+\eta_1} = \frac{\lambda}{1+\eta_2}$. Substituting into (30) gives

$$
\sum_{t=1}^{T} (H_t + M_t) \leq \frac{1+\eta}{\lambda} \sum_{t=1}^{T} \left( \hat{H}_t + \lambda \hat{M}_t \right) + \lambda \left( \frac{l}{1+\eta-\lambda} + \frac{4\alpha^2}{\eta} - \frac{m}{\lambda + m} \right) \cdot \sum_{t=1}^{T} \frac{1}{2} \|v_t - \tilde{v}_t\|^2 .
\tag{31}
$$

**Algorithm 4:** Optimistic ROBD with $\lambda = 0$

---

**for** $t = 1$ **to** $T$ **do**
  **Observe:** $v_{t-1}, h_t, \Omega_t$
  $s_t \leftarrow \sum_{i=1}^{p} C_i v_{-i}$
  Let $y_t$ be the projection of $s_t$ on $\Omega_t$
  **Output:** $y_t$ (the decision at time step $t$)

---

Recall that the point sequence $\{\hat{y}_t\}_{1 \leq t \leq T}$ is identical with the one picked by ROBD, which has parameters $\lambda_1 = \lambda, \lambda_2 = 0$ and has access to the exact $v_t$ before picking $\hat{y}_t$. Therefore, the same upper bound of $\sum_{t=1}^{T} \left( \hat{H}_t + \lambda \hat{M}_t \right)$ given in (9) in the proof of Theorem 1 also applies here. It shows that

$$\sum_{t=1}^{T} (\hat{H}_t + \lambda \hat{M}_t) \leq \sum_{t=1}^{T} \left( H_t^* + \frac{\lambda(\lambda + m)}{(1 - \alpha^2)\lambda + m} M_t^* \right). \tag{32}$$

Substituting (32) into (31) finishes the proof.

## F   Optimistic ROBD with $\lambda = 0$

Although Theorem 2 does cover the case when $\lambda = 0$, it is possible to extend the analysis to cover this setting. Notice that the agent may choose any point in $\Omega_t$ in Algorithm 2 when $\lambda = 0$. Thus, a tiebreaking rule is needed to cover the case of $\lambda = 0$. We break the tie by choosing the projection of $\sum_{i=1}^{p} C_i v_{t-i}$ on $\Omega_t$, which is natural if we consider $\lambda \to 0^+$. We give the pseudo for this specific case in Algorithm 4.

As in Section 3, we first consider the case when $\Omega_t$ is a one-point set, i.e. $\Omega_t = \{v_t\}$.

**Theorem 5.** *Suppose the hitting cost functions are $m-$strongly convex and the switching cost is given by $c(y_{t:t-p}) = \frac{1}{2} \left\| y_t - \sum_{i=1}^{p} C_i y_{t-i} \right\|_2^2$, where $C_i \in \mathbb{R}^{d \times d}$ and $\sum_{i=1}^{p} \|C_i\|_2 = \alpha$. When $\Omega_t = \{v_t\}$, the competitive ratio of Algorithm 4 is upper bounded by $1 + \frac{(1+\alpha)^2}{m}$.*

*Proof of Theorem 5.* Notice that when $\Omega_t = \{v_t\}$, Algorithm 4 will pick $y_t = v_t$ for all time step $t$. Since $v_t = \arg\min_y f_t(y)$ and $f_t$ is $m-$strongly convex, we have that

$$f_t(v_t) + \frac{m}{2} \|y_t^* - v_t\|^2 \leq f_t(y_t^*). \tag{33}$$

On the other hand, we can bound the switching cost of Algorithm 4 by

$$\frac{1}{2} \left\| v_t - \sum_{i=1}^{p} C_i v_{t-i} \right\|^2$$

$$= \frac{1}{2} \left\| y_t^* - \sum_{i=1}^{p} C_i y_{t-i}^* \right\|^2 + \left\langle y_t^* - \sum_{i=1}^{p} C_i y_{t-i}^*, v_t - \sum_{i=1}^{p} C_i v_{t-i} \right\rangle + \frac{1}{2} \left\| (v_t - y_t^*) - \sum_{i=1}^{p} C_i (v_{t-i} - y_{t-i}^*) \right\|^2$$

$$\leq \frac{1}{2} \left\| y_t^* - \sum_{i=1}^{p} C_i y_{t-i}^* \right\|^2 + \left\| y_t^* - \sum_{i=1}^{p} C_i y_{t-i}^* \right\| \cdot \left\| v_t - \sum_{i=1}^{p} C_i v_{t-i} \right\| + \frac{1}{2} \left\| (v_t - y_t^*) - \sum_{i=1}^{p} C_i (v_{t-i} - y_{t-i}^*) \right\|^2 \tag{34a}$$

$$\leq \left( 1 + \frac{(1+\alpha)^2}{m} \right) \cdot \frac{1}{2} \left\| y_t^* - \sum_{i=1}^{p} C_i y_{t-i}^* \right\|^2 + \left( 1 + \frac{m}{(1+\alpha)^2} \right) \cdot \frac{1}{2} \left\| (v_t - y_t^*) - \sum_{i=1}^{p} C_i (v_{t-i} - y_{t-i}^*) \right\|^2, \tag{34b}$$

where we use Cauchy–Schwartz inequality in (34a); we use AM-GM inequality in (34b).

Notice that

$$\left\| (v_t - y_t^*) - \sum_{i=1}^{p} C_i(v_{t-i} - y_{t-i}^*) \right\|^2 \leq \left( \|v_t - y_t^*\| + \sum_{i=1}^{p} \|C_i\| \cdot \|v_{t-i} - y_{t-i}^*\| \right)^2 \quad (35a)$$

$$\leq (1 + \alpha) \cdot \left( \|v_t - y_t^*\|^2 + \sum_{i=1}^{p} \|C_i\| \cdot \|v_{t-i} - y_{t-i}^*\|^2 \right), \quad (35b)$$

where we use the triangle inequality in (35a) and the Cauchy-Schwartz inequality in (35b).

Substituting (35) into (34) and summing up through time steps, we obtain that

$$\sum_{t=1}^{T} \frac{1}{2} \left\| v_t - \sum_{i=1}^{p} C_i v_{t-i} \right\|^2 \leq \sum_{t=1}^{T} \left( 1 + \frac{(1+\alpha)^2}{m} \right) M_t^* + ((1+\alpha)^2 + m) \cdot \frac{1}{2} \|v_t - y_t^*\|^2. \quad (36)$$

Substituting (33) gives that

$$\sum_{t=1}^{T} \frac{1}{2} \left\| v_t - \sum_{i=1}^{p} C_i v_{t-i} \right\|^2 \leq \sum_{t=1}^{T} \left( 1 + \frac{(1+\alpha)^2}{m} \right) M_t^* + \left( 1 + \frac{(1+\alpha)^2}{m} \right) \cdot (H_t^* - f_t(v_t)),$$

which implies

$$\sum_{t=1}^{T} \left( f_t(v_t) + \frac{1}{2} \left\| v_t - \sum_{i=1}^{p} C_i v_{t-i} \right\|^2 \right) \leq \left( 1 + \frac{(1+\alpha)^2}{m} \right) \sum_{t=1}^{T} (H_t^* + M_t^*). \quad (37)$$

$\square$

Now we consider the case when $\Omega_t$ is a general convex set.

**Theorem 6.** *Suppose the hitting cost functions are both $m-$strongly convex and $l-$strongly smooth and the switching cost is given by $c(y_{t:t-p}) = \frac{1}{2} \|y_t - \sum_{i=1}^{p} C_i y_{t-i}\|_2^2$, where $C_i \in \mathbb{R}^{d \times d}$ and $\sum_{i=1}^{p} \|C_i\|_2 = \alpha$. For arbitrary $\eta > 0$, the cost of Algorithm 4 is upper bounded by $K_1 \operatorname{cost}(OPT) + K_2$, where:*

$$K_1 = (1 + \eta) \cdot \left( 1 + \frac{(1+\alpha)^2}{m} \right),$$

$$K_2 = \left( l + \left( 1 + \frac{1}{\eta} \right) \alpha^2 - (1 + \eta) \right) \cdot \sum_{t=1}^{T} \frac{1}{2} \|y_t - v_t\|^2.$$

Like Theorem 2, we can choose $\eta$ to balance $K_1$ and $K_2$ and obtain a competitive ratio, in particular the smallest $\eta$ such that:

$$l + \left( 1 + \frac{1}{\eta} \right) \alpha^2 - (1 + \eta) \leq 0.$$

Therefore, we have $\eta = O(l + \alpha^2)$ and $K_2 \leq 0$. So the competitive ratio is upper bounded by:

$$O \left( (l + \alpha^2) \cdot \left( 1 + \frac{(1+\alpha)^2}{m} \right) \right).$$

*Proof of Theorem 6.* Since $y_t$ is the projection of $\sum_{i=1}^{p} C_i v_{t-i}$ on $\Omega_t$, and $\Omega_t$ is a convex set, we have that

$$\frac{1}{2} \left\| y_t - \sum_{i=1}^{p} C_i v_{t-i} \right\|^2 \leq \frac{1}{2} \left\| v_t - \sum_{i=1}^{p} C_i v_{t-i} \right\|^2 - \frac{1}{2} \|v_t - y_t\|^2. \quad (38)$$

Because the hitting cost function $f_t$ is $l$-strongly smooth, and $v_t$ is the minimizer of $f_t$, we see that

$$\frac{1}{\eta_1} f_t(y_t) \leq \frac{l}{2\eta_1} \|v_t - y_t\|^2 + \frac{1}{\eta_1} f_t(v_t) \quad (39)$$

holds for any $\eta_1 \geq 1$.

Since function $\frac{1}{2}\|y_t - y\|^2$ is 1-strongly smooth in $y$, by Lemma 4, we see that for any $\eta_2 > 0$,

$$\frac{1}{1+\eta_2} \cdot \frac{1}{2}\left\|y_t - \sum_{i=1}^{p} C_i y_{t-i}\right\|^2 \leq \frac{1}{2}\left\|y_t - \sum_{i=1}^{p} C_i v_{t-i}\right\|^2 + \frac{1}{2\eta_2}\left\|\sum_{i=1}^{p} C_i(v_{t-i} - y_{t-i})\right\|^2. \quad (40)$$

Notice that

$$\frac{1}{2}\left\|\sum_{i=1}^{p} C_i(v_{t-i} - y_{t-i})\right\|^2 \leq \frac{1}{2}\left(\sum_{i=1}^{p}\|C_i\| \cdot \|y_{t-i} - v_{t-i}\|\right)^2 \quad (41a)$$

$$\leq \frac{\alpha}{2}\left(\sum_{i=1}^{p}\|C_i\| \cdot \|y_{t-i} - v_{t-i}\|^2\right), \quad (41b)$$

where we use the triangle inequality and the definition of matrix norm in (41a); Jensen's Inequality in (41b).

Substituting (41) into (40) gives

$$\frac{1}{1+\eta_2} \cdot \frac{1}{2}\left\|y_t - \sum_{i=1}^{p} C_i y_{t-i}\right\|^2 \leq \frac{1}{2}\left\|y_t - \sum_{i=1}^{p} C_i v_{t-i}\right\|^2 + \frac{\alpha}{2\eta_2}\left(\sum_{i=1}^{p}\|C_i\| \cdot \|y_{t-i} - v_{t-i}\|^2\right). \quad (42)$$

Substituting (39) and (42) into (38) gives

$$\frac{1}{\eta_1}f_t(y_t) + \frac{1}{1+\eta_2} \cdot \frac{1}{2}\left\|y_t - \sum_{i=1}^{p} C_i y_{t-i}\right\|^2$$

$$\leq \frac{1}{\eta_1}f_t(v_t) + \frac{1}{2}\left\|v_t - \sum_{i=1}^{p} C_i v_{t-i}\right\|^2 + \left(\frac{l}{\eta_1} - 1\right) \cdot \frac{1}{2}\|v_t - y_t\|^2 + \frac{\alpha}{2\eta_2}\left(\sum_{i=1}^{p}\|C_i\| \cdot \|y_{t-i} - v_{t-i}\|^2\right). \quad (43)$$

Summing up (43) through time steps, we obtain that

$$\min\{\frac{1}{\eta_1}, \frac{1}{1+\eta_2}\}\sum_{t=1}^{T}\left(f_t(y_t) + \frac{1}{2}\left\|y_t - \sum_{i=1}^{p} C_i y_{t-i}\right\|^2\right)$$

$$\leq \sum_{t=1}^{T}\left(f_t(v_t) + \frac{1}{2}\left\|v_t - \sum_{i=1}^{p} C_i v_{t-i}\right\|^2\right) + \left(\frac{l}{\eta_1} + \frac{\alpha^2}{\eta_2} - 1\right) \cdot \frac{1}{2}\|y_t - v_t\|^2. \quad (44)$$

Let $\eta_2 = \eta$ and $\eta_1 = 1 + \eta$. Combining with (37), we obtain that

$$\sum_{t=1}^{T}\left(f_t(y_t) + \frac{1}{2}\left\|y_t - \sum_{i=1}^{p} C_i y_{t-i}\right\|^2\right)$$

$$\leq (1+\eta) \cdot \left(1 + \frac{(1+\alpha)^2}{m}\right) \cdot \sum_{t=1}^{T}(H_t^* + M_t^*) + \left(l + \left(1 + \frac{1}{\eta}\right)\alpha^2 - (1+\eta)\right) \cdot \frac{1}{2}\|y_t - v_t\|^2. \quad (45)$$

$\square$

## G  Proof and example of Theorem 3

The proof will proceed as follows. First, we extract the controllable dimensions in $x_t$, $\{x_t^{(k_1)}, \cdots, x_t^{(k_d)}\}$, to construct a new vector $z_t$. Then we can represent $x_t$ by $z_t, z_{t-1}, \cdots, z_{t-p}$.

Therefore, we can rewrite the dynamics in sequence $\{z_t\}_{0\le t\le T}$, control action $u_t$, and noise $w_t$. By this approach, we can remove the control matrix $B$ before $(u_t + w_t)$ in the dynamics. Finally, we can convert the resulting dynamics to an OCO problem with structured memory.

We use $\|\cdot\|$ to denote $\ell_2$ norm throughout the proof.

Recall that the objective is given as

$$\frac{1}{2} \sum_{t=0}^{T} \left( q_t \|x_t\|^2 + \|u_t\|^2 \right), \tag{46}$$

where $q_t > 0$ for all $0 \le t \le T$. Without loss of generality, we assume $q_t = 0$ for all $t > T$.

Recall that we define operator $\psi : \mathbb{R}^n \to \mathbb{R}^m$ as

$$\psi(x) = \left( x^{(k_1)}, \cdots, x^{(k_d)} \right)^{\mathsf{T}}.$$

Using this notation, we define vector $z_t$ as

$$z_t := \psi(x_t), t \ge 0.$$

Notice that $z_t^j = x_t^{(k_j)}$ for $j = 1, \cdots, d$. Since we have $x_t^{(i)} = x_{t-1}^{(i+1)}$ for $i \notin \mathcal{I}$, $x_t$ can be represented by

$$x_t = \left( z_{t-p_1+1}^{(1)}, \cdots, z_t^{(1)}, \cdots, z_{t-p_d+1}^{(d)}, \cdots, z_t^{(d)} \right)^{\mathsf{T}}. \tag{47}$$

Since $x_0 = \mathbf{0}$, we have $z_t = 0$ for $t \le 0$.

Using (47), we can rewrite the objective function as a function of sequence $\{z_t\}$ and $\{u_t\}$. Notice that

$$\sum_{t=0}^{T} q_t \|x_t\|_2^2 = \sum_{t=0}^{T} q_t \sum_{i=1}^{d} \sum_{j=1}^{p_i} \left( z_{t+1-j}^{(i)} \right)^2$$

$$= \sum_{t=0}^{T-1} \sum_{i=1}^{d} \left( \sum_{j=1}^{p_i} q_{t+j} \right) \left( z_{t+1}^{(i)} \right)^2, \tag{48a}$$

where in (48a) we use $z_t = \mathbf{0}$ for all $t \le 0$ and $q_t = 0$ for all $t > T$.

Therefore, we define function $h_t : \mathbb{R}^d \to \mathbb{R}^+ \cup \{0\}$ as

$$h_t(y) = \frac{1}{2} \sum_{i=1}^{d} \left( \sum_{j=1}^{p_i} q_{t+j} \right) \left( y^{(i)} \right)^2.$$

Using this definition, the objective (46) can be rewrite as

$$\frac{1}{2} \sum_{t=0}^{T} \left( q_t \|x_t\|^2 + \|u_t\|^2 \right) = \sum_{t=0}^{T-1} h_t(z_{t+1}) + \frac{1}{2} \|u_t\|^2, \tag{49}$$

where we notice that the optimal choice of control action $u_T$ is always zero because it will not affect any state.

We also see that $u_t$ can be determined by $z_{t-p+1:t+1}$ because

$$u_t = z_{t+1} - w_t - A(\mathcal{I}, :)x_t, \tag{50}$$

where $A(\mathcal{I}, :)$ consists of $k_1, \cdots, k_n$ rows of $A$ and $t \ge 0$.

Notice that $A(\mathcal{I}, :)x_t$ can be written as $\sum_{i=1}^{p} C_i z_{t-i+1}$ by the definition of $C_i, i = 1, \cdots, p$. Therefore, we can rewrite (50) as

$$u_t = z_{t+1} - w_t - \sum_{i=1}^{p} C_i z_{t-i+1}, \tag{51}$$

which is equivalent to

$$z_{t+1} = u_t + w_t + \sum_{i=1}^{p} C_i z_{t-i+1}.$$

We recursively define sequence $\{y_t\}_{t \geq -p}$ as the accumulation of control actions, i.e.

$$y_t = u_t + \sum_{i=1}^{p} C_i y_{t-i}, \forall t \geq 0,$$

where $y_t = \mathbf{0}$ for all $t < 0$. We also define sequence $\{\zeta_t\}_{t \geq -p}$ as the accumulation of control noises, i.e.

$$\zeta_t = w_t + \sum_{i=1}^{p} C_i \zeta_{t-i}, \forall t \geq 0,$$

where $\zeta_t = \mathbf{0}$ for all $t < 0$.

Recall that we have $x_0 = \mathbf{0}$ by assumption. Therefore,

$$z_{t+1} = y_t + \zeta_t \tag{52}$$

holds for all $t \geq -1$.

Using (49) and (52), we can formalize the problem as *online optimization with memory*, where the hitting cost function is given by

$$f_t(y) = h_t(y + \zeta_t),$$

and the switching cost is $\frac{1}{2} \|y_t - \sum_{i=1}^{p} C_i y_{t-i}\|^2$.

Although $h_t$ is revealed before the agent picks $y_t$, we need the knowledge of $v_t = -\zeta_t$ to construct the hitting cost function $f_t$, which depends on previous noises $w_{0:t}$. At time step $t$, we know the exact $w_\tau$ for all $\tau \leq t - 1$, thus we can compute the exact $\zeta_\tau$ for all $\tau \leq t - 1$. Since the set $W_t$ contains all possible noise $w_t$, we can construct the set $\Omega_t = \{-w - \sum_{i=1}^{p} C_i \zeta_{t-i} \mid w \in W_t\}$ which contains all possible $v_t$.

**Example.** To illustrate the reduction, consider the following example:

$$\begin{bmatrix} x_{t+1}^{(1)} \\ x_{t+1}^{(2)} \\ x_{t+1}^{(3)} \\ x_{t+1}^{(4)} \\ x_{t+1}^{(5)} \end{bmatrix} = \begin{bmatrix} 0 & 1 & 0 & 0 & 0 \\ a_1 & a_2 & a_3 & a_4 & a_5 \\ 0 & 0 & 0 & 1 & 0 \\ 0 & 0 & 0 & 0 & 1 \\ b_1 & b_2 & b_3 & b_4 & b_5 \end{bmatrix} \begin{bmatrix} x_t^{(1)} \\ x_t^{(2)} \\ x_t^{(3)} \\ x_t^{(4)} \\ x_t^{(5)} \end{bmatrix} + \begin{bmatrix} 0 & 0 \\ 1 & 0 \\ 0 & 0 \\ 0 & 0 \\ 0 & 1 \end{bmatrix} \left( \begin{bmatrix} u_t^{(1)} \\ u_t^{(2)} \end{bmatrix} + \begin{bmatrix} w_t^{(1)} \\ w_t^{(2)} \end{bmatrix} \right). \tag{53}$$

Notice that since $x_{t+1}^{(1)} = x_t^{(2)}, x_{t+1}^{(3)} = x_t^{(4)}$, we can rewrite (53) in a more compact form:

$$\underbrace{\begin{bmatrix} x_{t+1}^{(2)} \\ x_{t+1}^{(5)} \end{bmatrix}}_{z_{t+1}} = \underbrace{\begin{bmatrix} a_2 & a_5 \\ b_2 & b_5 \end{bmatrix}}_{C_1} \begin{bmatrix} x_t^{(2)} \\ x_t^{(5)} \end{bmatrix} + \underbrace{\begin{bmatrix} a_1 & a_4 \\ b_1 & b_4 \end{bmatrix}}_{C_2} \begin{bmatrix} x_{t-1}^{(2)} \\ x_{t-1}^{(5)} \end{bmatrix} + \underbrace{\begin{bmatrix} 0 & a_3 \\ 0 & b_3 \end{bmatrix}}_{C_3} \begin{bmatrix} x_{t-2}^{(2)} \\ x_{t-2}^{(5)} \end{bmatrix} + \begin{bmatrix} u_t^{(1)} \\ u_t^{(2)} \end{bmatrix} + \begin{bmatrix} w_t^{(1)} \\ w_t^{(2)} \end{bmatrix}. \tag{54}$$

In this example $p_1 = 2, p_2 = 3, \mathcal{I} = \{k_1, k_2\} = \{2, 5\}$ and thus $p = 3$ and $n = 2$. From (54) we have

$$z_{t+1} = C_1 z_t + C_2 z_{t-1} + C_3 z_{t-2} + u_t + w_t. \tag{55}$$

Recall the definition of $y_t$ and $\zeta_t$:

$$y_t = u_t + \sum_{i=1}^{3} C_i y_{t-i}, \forall t \geq 0, \quad \zeta_t = w_t + \sum_{i=1}^{3} C_i \zeta_{t-i}, \forall t \geq 0. \tag{56}$$

Then the original system could be translated to the compact form:

$$z_{t+1} = y_t + \zeta_t. \tag{57}$$

**Algorithm 5:** Adaptive control via optimistic ROBD

---

**Parameter:** $\lambda > 0$
**Input:** Transition matrix $A$ and control matrix $B$
**for** $t = 0$ **to** $T - 1$ **do**

    **Observe:** $x_t$, $W_t$, and $q_{t:t+p-1}$
    **if** $t > 0$ **then**
        $w_{t-1} \leftarrow \psi\left(x_t - Ax_{t-1} - Bu_{t-1}\right)$
        $\hat{z}_t \leftarrow \psi(x_t)$
    Define function $h_t(z) = \frac{1}{2} \sum_{i=1}^d \left( \sum_{j=1}^{p_i} q_{t+j} \right) \left(z^{(i)}\right)^2$
    $\tilde{w}_t \leftarrow \arg\min_{w \in W_t} \min_z h_t(z) + \frac{\lambda}{2} \left\| z - w - \sum_{i=1}^p C_i \hat{z}_{t+1-i} \right\|^2$
    $z_t \leftarrow \arg\min_z h_t(z) + \frac{\lambda}{2} \left\| z - \tilde{w}_t - \sum_{i=1}^p C_i \hat{z}_{t+1-i} \right\|^2$
    $u_t \leftarrow z_t - \tilde{w}_t - \sum_{i=1}^p C_i z_{t-i}$
    **Output:** $u_t$
**Output:** $u_T = 0$

---

If the objective is given as (46), we have that

$$h_t(z) = \frac{1}{2}(q_{t+1} + q_{t+2})\left(z^{(1)}\right)^2 + \frac{1}{2}(q_{t+1} + q_{t+2} + q_{t+3})\left(z^{(2)}\right)^2.$$

Lastly, we want to point out that our reduction can work for more general forms of objectives than (46). Specifically, we only require that the objective can be transformed to

$$\sum_{t=0}^{T-1} h_t(z_{t+1}) + \frac{1}{2}\|u_t\|^2,$$

where $h_t$ is a strongly convex and strongly smooth function that is observable before the agent picks $u_t$. Therefore, our reduction is more general than the reduction given in [24][Corollary 2], which considered the case when $B = I$. Notice that when $B = I$, we have $p = 1$ and $z_t = x_t$.

## H   A numerical issue in algorithm 3 and its solution

We have presented Algorithm 3 in as simple and intuitive a manner as possible but, as a result, there is a potential numerical issue that may arise for large horizon $T$. Although the sequence $\{z_t\}$ is naturally bounded and we always have $z_{t+1} = y_t + \zeta_t$, the magnitudes of $y_t$ and $\zeta_t$ may grow exponentially since they accumulate the actions and the noises separately. However, this is not a fundamental problem, and there is a straightforward solution when the *Solver* in Algorithm 3 is Optimistic ROBD (Algorithm 2). The key insight is to solve optimization in $\{u_t, w_t, z_t\}$ space, instead of $\{y_t, \zeta_t, z_t\}$ space.

More specifically, when instantiated with Optimistic ROBD, we can rewrite the pseudo code of Algorithm 3 as Algorithm 5 so that variables $y_t$ and $\zeta_t$ are not involved. While equivalent to Algorithm 3 with Optimistic ROBD as the *Solver*, Algorithm 5 is numerically stable because we avoid the potentially unstable recursive calculation of $\zeta_t$ and the sequence $\{w_t\}$ is bounded.

## I   Proofs for Appendix B

In this section, we establish the lower bound of the cost incurred by any linear controller and the upper bound of the offline optimal cost for different noise sequences. Specifically, we show a lower bound of the linear controller's cost on any noise sequence in Section I.1. We also give an upper bound of the offline optimal cost on any noise sequence in Section I.2. We further show that the upper bound of the offline optimal cost can be improved on two specific noise sequences in Section I.3 and I.4. Based on these results, we derive the lower bound of the competitive ratio for any linear control with respect to the these noise sequences in Section I.5, I.6, and I.7.

## I.1 Lower bound of $\mathtt{cost}(LC)$ for any noise sequence $\{w_t\}_{t=0}^T$

For any stable linear controller $u_t = -kx_t$, we have the following closed-loop dynamics

$$x_{t+1} = (a-k)x_t + w_t.$$

Our technique is to consider the sum of squares of two consecutive states $x_{t+1}$ and $x_t$. Due to the constraints given by the dynamics and the linear controller itself, $x_{t+1}$ and $x_t$ cannot reach zero simultaneously. Specifically, we define $\beta = a - k$. Since the controller is stable, we have $-1 < \beta < 1$. Consider $|x_{t+1}|^2 + |x_t|^2, \forall t \geq 0$, we have:

$$
\begin{aligned}
&|x_{t+1}|^2 + |x_t|^2 \\
=&(\beta x_t + w_t)^2 + x_t^2 \\
=&(\beta^2 + 1)x_t^2 + 2\beta x_t w_t + w_t^2 \\
=&(\beta^2 + 1)(x_t + \frac{\beta}{\beta^2 + 1}w_t)^2 + \frac{1}{\beta^2 + 1}w_t^2 \\
\geq&\frac{1}{\beta^2 + 1}w_t^2 > \frac{w_t^2}{2}.
\end{aligned}
$$

Since $\mathtt{cost}(LC) = \sum_{t=0}^T qx_t^2 + u_t^2 = \sum_{t=0}^T (q+k^2)x_t^2$, $\mathtt{cost}(LC) \geq \sum_{t=0}^{T-1}(q+k^2)x_{t+1}^2$. Then we will have

$$\mathtt{cost}(LC) \geq \frac{1}{2}\sum_{t=0}^{T-1}(q+k^2)(x_{t+1}^2 + x_t^2) > \frac{q+k^2}{4}\sum_{t=0}^{T-1}w_t^2 > \frac{q+(a-1)^2}{4}\sum_{t=0}^{T-1}w_t^2, \quad (58)$$

where the last step comes from the fact $-1 < a - k < 1$ and $a > 1$.

## I.2 Upper bound of $\mathtt{cost}(OPT)$ for any $\{w_t\}_{t=0}^T$

When the controller has the full knowledge of the future noise sequence, the simplest strategy is to correct the noise greedily at the start of each time step so that the agent always stays at state $0$.

Formally, for $\mathtt{cost}(OPT)$, consider controller $u_t = -w_t, \forall t \neq T$ and $u_t = 0, t = T$. Then we will have $x_t = 0, \forall t \leq T$ so the cost would be $\sum_{t=0}^{T-1} w_t^2$. Therefore we have

$$\mathtt{cost}(OPT) \leq \sum_{t=0}^{T-1} w_t^2.$$

## I.3 Upper bound of $\mathtt{cost}(OPT)$ for $w_t = w$

Compared with Section I.2, since $w_t$ is a constant case, we can balance the hitting cost and the switching cost by keeping the agent at non-zero stationary state that is close to the zero state.

Formally, we consider the following control strategy:

$$
u_t = \begin{cases} \frac{u+w}{1-a} - w, & t = 0 \\ u, & t \geq 1, \end{cases}
$$

where $u$ is another constant. This controller yields $x_t = \frac{u+w}{1-a}, t \geq 1$. Then, we have

$$\mathtt{cost}(u) = T(q(\frac{u+w}{1-a})^2 + u^2) + (\frac{u+w}{1-a} - w)^2,$$

where the first part is a quadratic function w.r.t. $u$ and the minimum is $\frac{q}{q+(a-1)^2} \cdot Tw^2$ with minimizer $u^* = \frac{-qw}{q+(a-1)^2}$. Therefore we get

$$\mathtt{cost}(OPT) \leq \frac{q}{q+(a-1)^2}Tw^2 + c_1,$$

where $c_1 = (\frac{u^*+w}{1-a} - w)^2$ is a constant.

## I.4 Upper bound of $\mathtt{cost}(OPT)$ for $w_t = (-1)^t \cdot w$

Instead of keeping the noise $w_t$ at a fixed value, we let it oscillate between two values $w$ and $-w$. The resulting offline optimal controller will also oscillate between a positive state and a negative state. We show that in this case, the offline optimal cost can be even smaller than the one when $w_t$ is fixed at $w$ (Section I.3).

In this case the dynamics follows

$$\begin{cases} x_{2k+1} = ax_{2k} + u_{2k} + w, & k \geq 0 \\ x_{2k+2} = ax_{2k+1} + u_{2k+1} - w, & k \geq 0. \end{cases}$$

Consider controller class

$$u_t = \begin{cases} -\frac{u-w}{a+1} - w, & t = 0 \\ u, & t = 2k+1, k \geq 0 \\ -u, & t = 2k+2, k \geq 0. \end{cases}$$

Following this controller class, we have

$$x_t = \begin{cases} -\frac{u-w}{a+1}, & t = 2k+1, k \geq 0 \\ \frac{u-w}{a+1}, & t = 2k+2, k \geq 0. \end{cases}$$

For simplicity, assume $T$ is an even number. Then, we have

$$\mathtt{cost}(u) = T(q(\frac{u-w}{a+1})^2 + u^2) + (\frac{u-w}{a+1} + w)^2.$$

Similarly, the first part of $\mathtt{cost}(u)$ is a quadratic function and the minimum is $\frac{q}{q+(a+1)^2} \cdot Tw^2$. Therefore, we have

$$\mathtt{cost}(OPT) \leq \frac{q}{q+(a+1)^2} Tw^2 + c_2,$$

where $c_2$ is also a constant.

## I.5 Lower bound of $\frac{\mathtt{cost}(LC)}{\mathtt{cost}(OPT)}$ for any $\{w_t\}_{t=0}^T$

Combining I.1 and I.2 we will have, for any $\{w_t\}_{t=0}^T$:

$$\frac{\mathtt{cost}(LC)}{\mathtt{cost}(OPT)} > \frac{\frac{q+(a-1)^2}{4} \sum_{t=0}^{T-1} w_t^2}{\sum_{t=0}^{T-1} w_t^2} = \frac{q+(a-1)^2}{4}.$$

## I.6 Lower bound of $\frac{\mathtt{cost}(LC)}{\mathtt{cost}(OPT)}$ for $w_t = w$

Combining I.1 and I.3, we will have, if $w_t = w$:

$$\frac{\mathtt{cost}(LC)}{\mathtt{cost}(OPT)} > \frac{\frac{q+(a-1)^2}{4} Tw^2}{\frac{q}{q+(a-1)^2} Tw^2 + c_1}.$$

Therefore as $T \to \infty$, $\frac{\mathtt{cost}(LC)}{\mathtt{cost}(OPT)} \geq \frac{q+(a-1)^2}{4} \cdot \frac{q+(a-1)^2}{q}$.

## I.7 Lower bound of $\frac{\mathtt{cost}(LC)}{\mathtt{cost}(OPT)}$ for $w_t = (-1)^t \cdot w$

Combining I.1 and I.4, we will have, if $w_t = (-1)^t \cdot w$:

$$\frac{\mathtt{cost}(LC)}{\mathtt{cost}(OPT)} > \frac{\frac{q+(a-1)^2}{4} Tw^2}{\frac{q}{q+(a+1)^2} Tw^2 + c_2}.$$

Therefore as $T \to \infty$, $\frac{\mathtt{cost}(LC)}{\mathtt{cost}(OPT)} \geq \frac{q+(a-1)^2}{4} \cdot \frac{q+(a+1)^2}{q}$.



[Supplementary Material 2]



(a) 1-d system, $w_t \sim U(-1, 1)$

Legend:
- Op-ROBD ($w_t$ known at $t$)
- Op-ROBD ($w_t$ unknown at $t$)
- LC
- OPT

y-axis: normalized cost
x-axis: $\lambda$