[Reviews · NeurIPS 2020]

Review 1

Summary and Contributions: - This paper considers "OCO with structured memory", which considers loss functions decomposable as an instantaneous hitting cost on the current action, plus a switching cost that's the squared 2-norm of the current action minus a known fixed linear combination C_i of past actions. - In this setting, the authors show matching upper and lower bounds for a competitive ratio of 1 + O((2-norm of C_i)^2), assuming that the losses are strongly convex. This uses the machinery of regularized online balanced descent, from [Goel et al. '19]. - An extension ("optimistic ROBD") is presented, to deal with the case where the decomposition of the loss feedback into switching+hitting costs is unknown. - The authors apply this framework to its motivating problem, a formulation of online control: isotropic LQR with input disturbances. A black-box reduction is formulated to the general structured memory problem, leading to a competitive ratio guarantee of the proposed online algorithm against the optimal offline control sequence.

Strengths: - This work acts a bridge between the well-established literature on OCO with switching costs and the ML community's recent focus on online/adaptive control. - The model for disturbances that the control application can handle is very general and expressive, and the offline comparator in the competitive ratio is very strong. - The concerns about the large factor of \alpha^2 in the competitive ratio is alleviated by the lower bound presented in Appendix E. Similarly, the discussion in Appendix B is a nice justification for why one might seek a competitive ratio against such a powerful comparator rather than linear controllers, even if one must suffer a large constant. - The "Optimistic ROBD" algorithm is interesting, independently of the control setting, using ROBD as a subroutine twice to obtain a lagged "accurate sequence", then to act on an optimistic model of the decomposition. I'd like to request a clarification about the novelty of this idea in the SOCO context; see the "Related work" section below.

Weaknesses: - The primary concern that dampens my excitement is that the reduction to control introduces many caveats: - The constants in the competitive ratio. First, even though the basic structured memory setting has a mild dependence on the C_i factor when the C_i represent operators like velocity and acceleration, in the reduction C_i contains columns of the dynamics matrix A, which needs to be in controllable canonical form. To transform a general matrix into this form, the coefficients in this matrix are the coefficients of the characteristic polynomial of A. For marginally stable systems with high eigenvalue multiplicity, this can incur constants as large as the exponential-in-p ones demonstrated in the equation after line 166. - The appearance of q_t's themselves in the competitive ratio also raises some confusion about whether which choices of adversarial dynamics might lead to vacuous competitive ratios. However, the ability to handle adversarial disturbances is unaffected by this concern, and interesting enough independently. - Foremost, the restrictiveness of the setting (isotropic loss function centered at 0, must be initialized at 0) vs. general LQR means that there is still work to be done before there is a fully satisfying connection between this SOCO/OBD literature and control. - Overall, I don't consider these to be deal-breakers; the paper is competing with a new and some what However, it would be useful to contrast the various assumptions with those made in the other cited works on competitive control.

Correctness: - The proofs appear to be correct.

Clarity: - The presentation and structure of the paper are very clear and user-friendly.

Relation to Prior Work: - The paper adequately cites prior work in the disparate domains it bridges. The discussion of offline competitive ratio vs. policy regret, with the example of Appendix B, is helpful to contrast this with existing solution concepts in online control. Out of curiosity, perhaps another natural comparison would be classical robust (i.e. H_\infty) control: is the competitive ratio of such a pessimistic controller also unbounded? - I could not tell from my own literature search (and the paper seems not to indicate) whether the algorithmic idea from optimistic ROBD is new; the problem setting of SOCO where the hitting+switching cost decomposition is unknown a priori should stand independently. The paper is currently unclear about this, citing no other work in the section about that algorithm, and only stating that this is the "first algorithm with a constant competitive ratio for online optimization with memory longer than one step". If this is indeed unprecedented, I would suggest highlighting this as a standalone contribution. Could the authors please clarify?

Reproducibility: Yes

Additional Feedback: *** post-response *** Thanks for the thoughtful response. Regarding the second point, my objection is not at all with dependences on q_max in general; I think there's a distinction between having competitive ratios include q_max vs. regret bounds (like the cited [31]). In the latter, it's just a scaling factor, while in the former, the additive gap between the cost and the optimal could scale as q_max^2. However, the discussion on infinite competitive ratios is enlightening. My overall evaluation remains the same, given that the control setting is rather restrictive, and there are multiple dependences on large factors in the CR.


Review 2

Summary and Contributions: This paper considers a more generalized smoothed online convex optimization (SOCO) setting, where in each round t the switching cost function depends on multiple previous actions instead of one action, and the hitting function is only partially revealed to the learner before picking the action. The authors proposed a novel algorithm called optimistic ROBD to solve this problem, which is proved to enjoy a constant, dimension-free competitive ratio wrt the dynamic offline optimal sequence. The authors also show that the proposed algorithm can be applied to a certain class of online control problems.

Strengths: 1. The problem is well-motivated. Most of the previous smooth online convex optimization models assume that in each round t the loss function is revealed before the action is picked, which is sometimes unrealistic and hinders their applications to many domains. This paper partially solves this problem by proposing a new model where the loss functions are revealed to the learner in two separate steps, which is novel and very interesting. The new setting also contain a more generalized cost function which is related to multiple previous actions. 2. There are two main theoretical results. The authors firstly make a straightforward extension of the classic ROBD algorithm to support the new switching cost function, then propose a novel optimistic mechanism to deal with the uncertainty of the hitting cost functions. The two methods successfully solve the proposed problems, and proved to enjoy constant competitive ratios.

Weaknesses: The main theorem (Theorem 2) only holds for strongly convex and smooth functions, which seems very limited. Previous work (e.g., [24]) only assume the loss functions are strongly convex (but not necessarily smooth), which is already a very restrictive assumption. It would be nice if the authors could give more examples of loss functions (other than quadratic functions) which satisfy these assumptions.

Correctness: I have read the main paper and made high level checks of the proofs, and I didn’t find any significant errors.

Clarity: The paper is generally well-written and structured clearly. I found Section 4 is a little difficult to follow, and it would be better if the authors can provide more background informaton of the online control settings and a mord detailed comparison to [23].

Relation to Prior Work: The relation to prior work is clearly discuessed in general. However: 1. It seems that the proof of Theorem 1 is largely based on the proof of Theorem 4 in [23]. It would be better if the authors can make it more clear in the main paper and the appendix. 2. As mentioned above, it would be better if the authors can provide more background informaton of the online control settings and a mord detailed comparison to [23].

Reproducibility: Yes

Additional Feedback: In line 207, for lamda>0, it is not very straightforward to see why eta=O(ell+alpha^2) can make the inequality hold. Can the authors provide more details? In line 153, I think the competitive ratio is of order O(1/sqrt{m}) instead of O(1/m). In line 150, to make all the conditions hold, it seems that we have to assume alpha^2<m+1.


Review 3

Summary and Contributions: This paper proposes a novel OCO with structured memory setting, and gives an algorithm Optimistic ROBD that achieves a dimension-free competitive ratio under this setting. As an application, the paper presents a reduction from an online control problem to OCO with structured memory, which generalizes previously known control settings with competitive policies.

Strengths: -The technical claims are sound and well written. The authors included comprehensive results. -The setting is novel, and is indeed a generalization of the SOCO setting.

Weaknesses: - It will be helpful if the authors could further support their assumptions with examples of control systems, or connections to classical control literature. Is this a new control setting? Under what control settings does it make sense to assume that future costs and noises are known? This is also related to my question about the interaction model in OCO with structured memory, see additional feedback. -In Theorem 2, the competitive ratio can be upper bounded in two ways: 1) a competitive ratio with an additive term; 2) a competitive ratio with a factor of (l + \alpha^2). In the first case, the additive term has the sum of \|v_t - \tilde{v}_t\|^2 over t, which can be very large. In the second case, how should I reconcile this factor with the counterexample in Appendix B, where the lipschitz constant also appears?

Correctness: Yes.

Clarity: The paper is well written.

Relation to Prior Work: Yes.

Reproducibility: Yes

Additional Feedback: -It would be helpful if the authors could justify the assumption in the interaction model, where the geometry of f_t is revealed before the agent picks y_t. -In the reduction, it is assumed that q is known ahead of time for p steps. What is a control scenario when this is true? -Is there a motivating example for using the “beyond worst case” result? When are the estimations accurate enough? -Typos in Appendix: Page 14, definition of g_t, should be \lambda instead of \lambda/2 for the term involving M_t? Page 16 equation (7), the minus term on the left hand side should be summed over i instead of t? And what is q?


Review 4

Summary and Contributions: This paper extends the smoothed online convex optimization problem setting to a more general setting of OCO with structured memory. This setting extends SOCO to include cost functions which are dependent on several steps of history, and which are hidden from the decision maker at decision time. They also provide a new algorithm based on Regularized Online Balanced Descent which has a constant competitive ratio in the new setting. After the development and analysis of the new algorithm, the authors reduce Input-Disturbed Squared Regulators (a class of online control problems) to OCO with structured memory. With this reduction, O-ROBD can be applied directly to these problems which is explored by the authors. Analytical and Numerical examples are provided in the appendix.

Strengths: This is an strong paper which makes significant contributions to understanding online control using online optimization with memory. The theory is well written and understandable from an outside perspective, and the assumptions are laid out clearly. The authors also seem to consistently go out of their way to include extensive background and context for their contributions to make reading the paper a pleasant experience.

Weaknesses: I understand that space is limited, but I feel the numerical examples provide a nice grounding for theoretical work. I think this is especially important as it is a key part of your motivation (i.e. showing that the optimal linear controller can have large cost compared to the optimal) in not following the usual regret bounds compared to the optimal linear controller. Other than this single comment, I don't have issue with the paper as currently presented.

Correctness: I only rigorously checked a portion of the theory presented in the paper and appendix for correctness. But given how the theory is presented (i.e. explicitly detailing how each step is made), I'm relatively confident in the authors ability.

Clarity: The paper is well written, and provides enough context for an outside reader to understand the contributions. I especially like the layout and approach to the appendix, where things are clearly laid out and used concepts are shown explicitly.

Relation to Prior Work: This work is well situated in the literature, and well motivated.

Reproducibility: Yes

Additional Feedback: === post-rebuttal == I thank the authors for going into detail for the other reviewers issues. Even with the concerns raised by the other reviewers, I'm quite happy with the paper. I would suggest expanding a bit on the assumptions you make and how these constrain the settings you care about, specifically giving some examples in the control regime which follow your assumptions. === pre-rebuttal === I would like to see two additions to make digesting the theoretical contributions easier. 1. A table of variables. For example, the variables H_t and M_t are defined in Appendix A, and used in subsequent appendices. But when trying to remember where they were first defined when seeing them used (i.e. appendix D) I immediately jumped to the statement of the theorem in the main text. By have a central location where most variables are defined this will make the paper much more readable. 2. Restatement of theorems in appendix before proofs. Again, this is a readability issue and not a part of my decision to accept or reject the paper.

[Author Response · NeurIPS 2020]

**Reviewer 1:** Thank you for your constructive feedback. We hope our response below addresses your concerns:

• Indeed the constant $\alpha$ in the CR bound could exponentially increase as $p$ increases, especially for marginally-stable systems and highly unstable systems. But we believe it is mainly from the system's property, instead of our algorithm's limitation. In other words, marginally stable or highly unstable $A$ matrices are inherently harder to achieve small CR. As you mentioned, any online policy will suffer $\alpha^2$ in the competitive ratio (Appendix E). It is interesting to understand which system will make online control harder (we partially discussed that in Appendix B, focusing on 1-D systems), and we will add more discussion about this point in the main paper.

• In online learning, it is common to have $q_{\min}$ and $q_{\max}$ in the CR or regret bounds (e.g.,[23, 31]). We also want to point out that even if $q$ is fixed, bounding competitive ratio is non-trivial, and often it is unbounded [7].

• We agree our control setting is a subset of general LQR. We emphasize that our goal in the control part is to show the possibility for a policy to be competitive to the true, dynamic optimal offline cost, something not shown before. The only prior result is [23], which uses invertible $B$ and known $w_t$ at time $t$. This goal is important because all other prior works focus on achieving small regret compared to the best linear policy. Given our example in Appendix B, one may wonder if it is even possible to match the offline optimal. We give a positive answer. Though our setting is not fully general, it strictly generalized the prior art. We see this as a significant step towards competitive control.

• The most significant difference between our approach and the classical robust control (e.g., $H_\infty$) is that $H_\infty$ is neither online nor adaptive, i.e., the policy will not change even if some $w_t$ is not adversarial. Our framework is naturally adaptive from the estimation set $W_t$ (i.e., better estimation leads to more aggressive policy). We will add this discussion.

• Being optimistic is a common and powerful heuristic in general online learning, especially in regret minimization (e.g., UCB in multi-armed bandit, efficient Q-learning). But our optimistic idea is quite novel in SOCO, naturally because previous settings focus on precise information cases and then there is no need for being optimistic.

**Reviewer 2:** Thank you for your constructive feedback. We hope our response below addresses your concerns:

•We feel our assumption on strongly convexity and smoothness is not a significant weakness because: (1) Assuming strongly convexity and smoothness are very common in the online learning and optimization community [30, 31]; (2) Strong smoothness is not needed in [24] because the agent has the perfect prediction of the next hitting cost function, but it is critical in our setting where the prediction is imperfect, because the competitive ratio can be unbounded otherwise (e.g., consider the case when the hitting cost is an indicator function).

• Our setting strictly generalizes [23], where $B$ is invertible and $w_t$ is perfectly known at step $t$. To address the clarity issues, we will add more background and make notations easier to follow (e.g., add a notation list).

• For technical questions: (1) In line 207, we need to make an additional assumption that $\alpha$ is large (strictly speaking, larger than a constant $c$ such that $c > 1$.) In this case, $\lambda$ must be in the order of $O(m)$. (2) In line 153, when $m$ tends to zero, the competitive ratio is of order $O(1/\sqrt{m})$ when $\alpha = 1$, and $O(1/m)$ if $\alpha > 1$. Hence we report $O(1/m)$ which is more conservative. (3) In line 150, it is not necessary to assume $\alpha^2 < m + 1$, because all assumptions in line 150 hold if we let $\lambda_2 = 0$ and $\lambda_1 = 2m/(m + \alpha^2 - 1 + \sqrt{(m + \alpha^2 - 1)^2 + 4m})$. We will discuss them in the revision.

**Reviewer 3:** Thank you for your constructive feedback. We hope our response below addresses your concerns:

• Previous competitive ratio results in SOCO focus on the setting when both the geometry and minimizer of $f_t$ are revealed before the agent picks $y_t$ [16, 23, 24]. In contrast, the minimizer is unknown when picking $y_t$ in our setting. Our setting generalizes the previous ones, and is practical in many cases. For example, in power systems the geometry is governed by network topology (usually revealed before decision making) and minimizer is decided by users (which could be revealed afterwards). When reduced to control, the geometry is from cost functions and the minimizer is from adversarial disturbance. It is why we need $p$ steps of future costs, but don't need any future disturbances. We want to emphasize (1) the access to future cost functions is common in control if the focus is on disturbances in dynamics (e.g., in LQ tracking problem the cost functions are pregiven) and (2) the only existing competitive policy [23] needs both future costs and disturbances, and we show a competitive policy exists even if disturbance is unknown in advance.

• Note that the cost bound in our Theorem 2 is $C_1 \cdot \text{cost(opt)} + (a + b - d) \sum_t \|v_t - \tilde{v}_t\|^2$, and we can get a pure competitive ratio $C_2 \cdot \text{cost(opt)}$ because we have a "$-d$" term before the path length $\sum_t \|v_t - \tilde{v}_t\|^2$. We name it "beyond worst case" because the first bound will be tighter if the estimation is more precise. But note that we can always get a constant-competitive result even if the estimation is totally off, and $C_2$ does not depend on $\sum_t \|v_t - \tilde{v}_t\|^2$. Our numerical examples in Appendix C provide the example: if $w_t$ is smooth (i.e., $\|w_{t+1} - w_t\|$ is small), then the path length is small and the first bound gets tighter (Fig. 1(b,d)). However, if the estimation set is large, the path length may get large and then the second bound would be better.

• For technical questions, we think the definition of $g_t$ on page 14 is correct because the switching cost has the coefficient of $1/2$ by definition (between line 109 and 110). In Eq. (7), we should change $t$ to $i$ and $q$ to $p$.

**Reviewer 4:** Thank you for the feedback on paper organization and presentation. We would move our examples in the main body (we will have one more page in the revision), add a notion table and restate all claims before the proofs.

[Meta-Review · NeurIPS 2020]

The reviewers provided a detailed review of the paper and were overall unanimous in their agreement towards accepting the paper. The reviewers certainly appreciate the link between OCO with switching costs and the recent online/adaptive control work in ML. I would strongly encourage the authors to take a closer look at the updated reviews and the suggestions provided for improvement.